# Upstream Remotely-Sensed Hydrological Variables and Their Standardization for Surface Runoff Reconstruction and Estimation of the Entire Mekong River Basin

**Linghao Zhou [1,2], Hok Sum Fok [1,2,*], Zhongtian Ma [1,2] and Qiang Chen [3]**

[1] School of Geodesy and Geomatics, Wuhan University, Wuhan 430079, China; lhzhou2016@whu.edu.cn (L.Z.); zt__ma@whu.edu.cn (Z.M.)

[2] Key Laboratory of Geospace Environment and Geodesy, Ministry of Education, Wuhan University, Wuhan 430079, China

[3] Geophysics Laboratory, Faculty of Science, Technology and Communication, University of Luxembourg, 2, avenue de l'Université, Esch-sur-Alzette, L-4365 Luxembourg, Luxembourg; qiang.chen@uni.lu

\* Correspondence: xshhuo@sgg.whu.edu.cn; Tel.: +86-027-6877-8649

**Abstract:** River water discharge (WD) is an essential component when monitoring a regional hydrological cycle. It is expressed in terms of surface runoff ($R$) when a unit of river basin surface area is considered. To compensate for the decreasing number of hydrological stations, remotely-sensed WD estimation has been widely promoted over the past two decades, due to its global coverage. Previously, remotely-sensed WD was reconstructed either by correlating nearby remotely-sensed surface responses (e.g., indices and hydraulic variables) with ground-based WD observations or by applying water balance formulations, in terms of $R$, over an entire river basin, assisted by hydrological modeling data. In contrast, the feasibility of using remotely-sensed hydrological variables (RSHVs) and their standardized forms together with water balance representations (WBR) obtained from the river upstream to reconstruct estuarine $R$ for an entire basin, has been rarely investigated. Therefore, our study aimed to construct a correlative relationship between the estuarine observed $R$ and the upstream, spatially averaged RSHVs, together with their standardized forms and WBR, for the Mekong River basin, using estuarine $R$ reconstructions, at a monthly temporal scale. We found that the reconstructed $R$ derived from the upstream, spatially averaged RSHVs agreed well with the observed $R$, which was also comparable to that calculated using traditional remote sensing data (RSD). Better performance was achieved using spatially averaged, standardized RSHVs, which should be potentially attributable to spatially integrated information and the ability to partly bypass systematic biases by both human (e.g., dam operation) and environmental effects in a standardized form. Comparison of the $R$ reconstructed using the upstream, spatially averaged, standardized RSHVs with that reconstructed from the traditional RSD, against the observed $R$, revealed a Pearson correlation coefficient (PCC) above 0.91 and below 0.81, a root-mean-squares error (RMSE) below 6.1 mm and above 8.5 mm, and a Nash–Sutcliffe model efficiency coefficient (NSE) above 0.823 and below 0.657, respectively. In terms of the standardized water balance representation (SWBR), the reconstructed $R$ yielded the best performance, with a PCC above 0.92, an RMSE below 5.9 mm, and an NSE above 0.838. External assessment demonstrated similar results. This finding indicated that the standardized RSHVs, in particular its water balance representations, could lead to further improvement in estuarine $R$ reconstructions for river basins affected by various systematic influences. Comparison between hydrological stations at the Mekong River Delta entrance and near the estuary mouth revealed tidally-induced backwater effects on the estimated $R$, with an RMSE difference of 4–5 mm (equivalent to 9–11% relative error).

**Keywords:** runoff; water balance standardization; GRACE satellite gravimetry; remote sensing hydrology; Mekong River Basin

---

## 1. Introduction

River water discharge (WD) is an important water balance component of the hydrological cycle. It is defined as the water volume rate passing through a cross-section of a river and is expressed in terms of surface runoff ($R$) when a unit of surface area over a river basin is considered [1]. Its monitoring is essential for increased water management efficiency and tracking the regional hydrological extremes (i.e., droughts and floods) that cause unpredictable losses to agriculture and economy [2–5]. Thus, continuous WD time series data are necessary within a river basin.

Traditionally, WD has been directly measured at hydrological stations along a river. However, due to the varied politics, economies, and topography of the countries along a river [6], the spatial distribution of hydrological stations is uneven and sparse. In addition, acquisition of discharge data and their entry into a database has been declining since the late 1970s [7]. A lack of funding to continuously operate and maintain WD data systems is one of the reasons for this decline, therefore, new, indirect methods for observing WD are gaining increased attention.

Several research studies have investigated potential remotely sensed WD estimation, based on recent advances in remote sensing (RS). RS techniques can be divided into two types, passive and active. Moderate Resolution Imaging Spectrometer (MODIS), Landsat Thematic Mapper (TM), and Enhanced TM Plus (ETM+) are examples of passive RS techniques, which measure instantaneous surface response parameters, such as water surface area, floodplain inundation, the Normalized Difference Vegetation Index (NDVI), and Land Surface Temperature (LST) [8–11]. These passive remote sensing data (RSD) have been directly correlated with water level (WL) or WD [12–14] and applied to several river basins over the past two decades [15–17]. However, these data are not hydrological quantities with direct causal relationships to WD.

As well as these passive RS quantities, remotely sensed hydraulic variables have been used as inputs to infer WD through hydraulic geometry (e.g., [18,19]). For instance, WD can be inferred by integrating remotely-sensed hydraulic variables with topographic information (e.g., DEM), through well-known hydraulic models, such as Manning's equation (e.g., [20–24]). An innovative functional relationship between discharge and river width has been established recently, using hydraulic geometry for WD estimation [25,26]. Novel procedures that optimize the unknown parameters in the modified Manning's equation have also been developed [27]. Nevertheless, the accuracy of the estimated WD is region-dependent, relating to detection ability with respect to small changes in river width [28] and the availability of roughness coefficients for some regions [29,30].

Satellite radar altimetry is one of the active RS techniques that can directly measure water level (WL) fluctuations over inland lakes and rivers (e.g., [31,32]). Since WL is a power function of WD (e.g., [33,34]), active, remotely sensed WD mainly correlates the measured satellite altimetric WL of a river with ground-based WD measurements (e.g., [35]). In addition to estimating WD using satellite ground tracks, a basin-wide WD estimation for the Mekong River Basin (MRB) has been demonstrated, through multi-satellite altimetric (e.g., ERS-2 and Envisat) WL data [36]. To improve the temporal resolution of WD estimation, multi-altimetry data have been assimilated into hydrodynamic (e.g., [37]) or dynamic stochastic process models, using Kalman filtering to estimate WD [38]. In addition, WD estimation based on satellite altimetric sea level anomaly data near the estuary has been demonstrated [39]. However, the accuracy of the satellite radar altimetry technique is limited by the size of its footprint (i.e., 5 km, 20 km, and 3.4 km for TOPEX, ERS-2, and Envisat satellites, respectively) and because radar signals are contaminated by land surfaces near the riverbank when river width is small (e.g., [40]).

The Gravity Recovery and Climate Experiment (GRACE) [41] is another active RS technique that has used large-scale, time-variable gravity changes to infer monthly terrestrial water storage

(TWS) fluctuations (e.g., [42,43]). GRACE TWS is useful for capturing large-scale, seasonal surface water changes, which in turn allows monitoring of the global hydrological cycle and its extremes (e.g., [44,45]). Since WD is a power function of GRACE TWS ($S$) (e.g., [46,47]), WD can be inferred from $S$. Another approach to infer WD is based on the water balance equation (i.e., $R = P - ET - \Delta S$) [48–50], where WD is in the form of surface runoff ($R$) to be consistent with the units of precipitation ($P$), evapotranspiration ($ET$), and water storage change ($\Delta S$). In addition, parameters $S$ and $\Delta S$ are useful for inferring both precipitation ($P$) (e.g., [51]) and evapotranspiration ($ET$) (e.g., [52,53]).

These hydrological variables—$P$, $ET$, $S$, and $R$—are the four water balance components. $P$, $ET$, and $S$ can be obtained from Tropical Rainfall Measuring Mission (TRMM), MODIS, and GRACE, respectively. These variables are referred to as remotely-sensed hydrological variables (RSHVs), while $R$ can only be indirectly inferred. In essence, remotely-sensed $P$, $ET$, and $S$ should have a direct causal relationship with $R$, and their standardized forms, which can be obtained through the subtraction of the data time series from the corresponding mean, divided by the corresponding standard deviation, should make their averaged time series more representative of regional characteristics [54]. This is particularly the case across regions with large differences in means and variances [55,56], reducing the systematic environmental influences [57]. RSHVs and their standardized forms are presumably better at capturing variations of $R$ and standardized $R$, respectively, through direct correlative analysis, though standardized hydrological variables have been traditionally applied to drought characterization [58,59]. Since $R$ inferred from the water balance equation is achieved through subtraction among the RSHVs, systematic biases due to both human (e.g., dam operation) and environmental influences should be reduced, and thus, the inferred $R$ from the water balance equation could be expected to outperform the correlative analysis of each individual RSHV, not to mention the inferred $R$ from the standardization of the water balance equation (i.e., $SRI = SPI - SETI - \Delta SI$). These two forms are referred to as water balance representations (WBR).

The Mekong River Delta (MRD), close to the estuary mouth where there are several river mouths, is crucial for the food (e.g., fish and agriculture products [60]) and water (e.g., [61]) security of Southeast Asia, making it an attractive geographic study region. However, the upper Mekong River Basin (MRB) has been undergoing massive hydropower development since the 1990s, with the aim of regulating $R$ during severe flood and drought periods [62,63]. As a consequence, modification of the upstream $R$ due to dam operation could well pose severe downstream impacts at different times and seasons of the year (e.g., [64–66]), extending down to the MRD. In essence, the effect of dam operation in the upstream MRB propagates and accumulates with the effects of other dams along the mainstream Mekong River until reaching the MRD, where the accumulated effect detected in the MRD should be approximately similar for any specific month, year-on-year [66]. This accumulated effect, including ice and snowmelt water in the upper MRB [67], should be partly systematic, based on the dam operation principle of increasing (decreasing) discharge in dry (wet) seasons, with little overall change to annual flow.

These reasons support our investigation into the feasibility of using RSHVs, their standardized forms, as well as water-balance representations (WBR) obtained from the upstream MRB to reconstruct time series of $R$ in the MRD at a monthly temporal scale. We can proceed on this basis. since the above standardization procedure (i.e., $R_{i,j} - median(R_j)$), and subtraction in WBR (i.e., $S_{i+1,j} - S_{i,j}$), can mitigate the systematic bias introduced by dam operations and other environmental influences. The reconstructed relationship can be externally validated by estimating the time series of $R$ at other locations in the MRD, with in situ station time series used for performance assessment. The commonly available RSD (i.e., NDVI [68] and LST [69]) can also serve as baseline results for comparison purposes.

The paper has been organized as follows: In Section 2, the geography of the MRB and of Yunnan Province in China have been presented. Datasets are described in Section 3, which is followed by methodology and evaluation metric descriptions in Section 4. In Section 5, reconstruction and estimation of $R$ based on RSHVs, their standardized forms, and WBR, have been investigated and then compared with NDVI and LST results. Finally, conclusions have been given in Section 6.

## 2. The Geography of the Mekong River Basin (MRB) and of Yunnan Province

The Mekong River, originating from the northeast slopes of the Tanggula Mountains, China, is the 12th longest river in the world, with a main stream total length of 4909 km [70]. The river first flows as the Lancang River (also called the Mekong River outside of China) in Yunnan Province, China, before making its way through Laos, Myanmar, Thailand, Cambodia, and Vietnam. These six countries constitute the entire MRB, which covers an area of ~795,000 km$^2$, spanning 25° of latitude [71] (Figure 1).

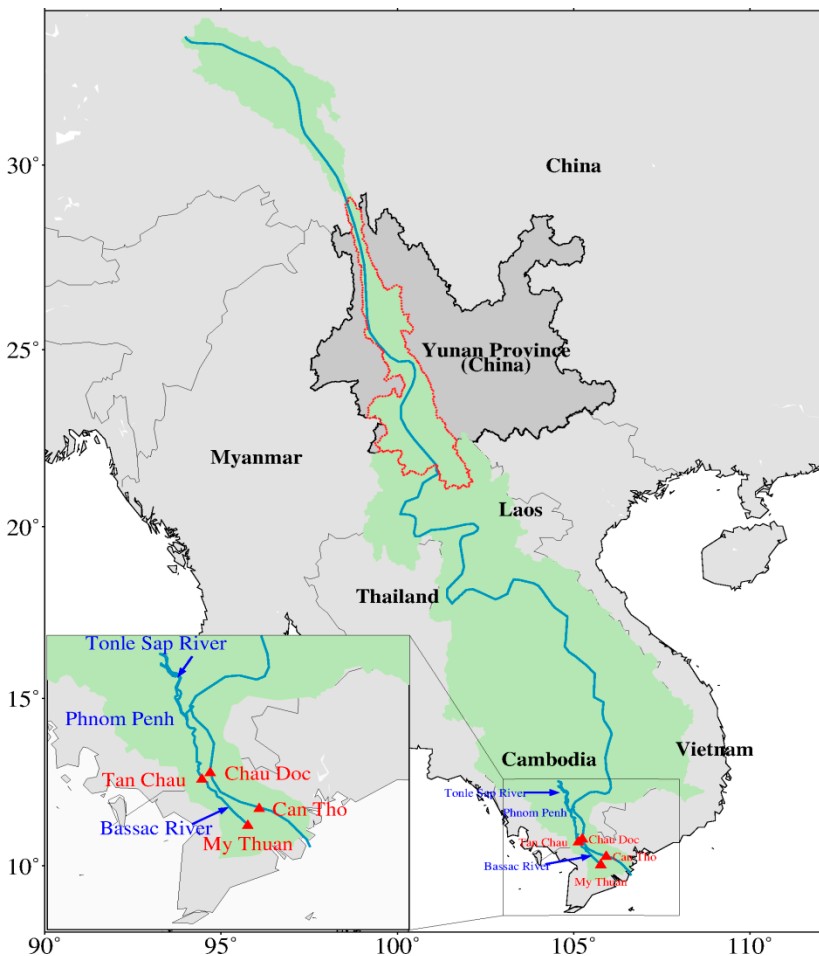

**Figure 1.** Map of the Mekong River Basin with two pairs of discharge gauge stations (My Thuan and Can Tho, and Tan Chau and Chau Doc) situated near the estuary mouth. Lancang River Basin boundary shown in red.

Located at the center of the Asian tropical monsoon region, the MRB, particularly in its lower reaches, is significantly influenced by the Asia–Pacific monsoon system [72]. The MRB rainy season extends from May to October and is controlled by the Southwest (SW) monsoon from the ocean, while the dry season, which lasts from November to April, is greatly influenced by the Northeast (NE) monsoon from the mainland [73]. It is noted that, apart from a small portion of ice and snowmelt water flowing from upstream, rainfall is the dominant $R$ component within the basin [74]. The water cycle consists of downstream $R$ rising abruptly in May, before peaking in late September during the rainy season, then decreasing from November and reaching its lowest level in April during the dry season [73], illustrating that $R$ for the entire MRB is greatly affected by monsoon behavior.

Yunnan Province, with mountainous plateau topography descending stepwise from north to south [75], is the main upstream MRB area. It is located in SW China, with an extent of 21°–29° N

latitude, and 97°–106° E longitude (Figure 1). It has an area of 394,100 km$^2$ and is inhabited by 47.7 million people [76]. It has a subtropical plateau monsoon climate that is jointly influenced by the SW and SE warm, moist air flows. Precipitation is distributed non-uniformly across the province, both in terms of space and time. The rainy and dry seasons are distinct, with the former (May–October) accounting for 85% of the annual rainfall [77].

## 3. Datasets

### 3.1. In Situ Hydrological Stations and Traditional Remotely-Sensed Data

Based on its discharge variability, the geographic setting of the MRD study site belongs to a fluvial to marine transition zone, where it is dynamically affected by a combination of fluvial and marine (i.e., tide and wave) processes that varies seasonally [78,79]. It is linked to a complicated river system, where the total discharge of the MRB is regulated by Cambodia's Tonle Sap Lake (e.g., [80–82]) before delivering to the MRD and discharging to the South China Sea (SCS) through several distributaries associated with the two main branches, the Bassac River and the Mekong River. This system is complicated by oceanic tidal propagation landward from the estuary mouth [60], and consequently, selection of the most appropriate hydrological stations is critical.

Given the above setting, the MRD hydrological stations to be used in this study had to be selected at locations where the regulating effects of Tonle Sap Lake and the ocean tides close to estuary were minimized. The Chaktomuk station at Phnom Penh is ~300 km away from the estuary mouth, at the intersection of several main distributaries. When river flows from where several main distributaries meet, the overall temporal pattern of discharge of the MRB mainstream could be altered, and so this station was not selected.

Given the above criteria, Tan Chau and Chau Doc, located at the MRD entrance, approximately 220 km from the estuary mouth and on the edge of the fluvial to marine transition zone [79], were considered to be the most appropriate hydrological stations. They were considered to be far enough away from the river mouth for the ocean tidal propagation effect to be reduced (Figure 1). Note that semi-diurnal (i.e., half-daily period) ocean tides are dominant in the SCS, and that ocean tidal influence can be mitigated by adopting a monthly averaging process. Therefore, the two stations above were chosen in this study. The Can Tho and My Thuan stations, closest to the estuary mouth, were also chosen to assess the impact of the backwater effect, caused by landward ocean tidal propagation, on surface runoff estimation.

Station data were obtained from the Mekong River Commission (MRC) (http://www.mrcmekong.org). In order to reduce the effect of newly constructed dams in Yunnan Province (e.g., Nuozhadu Dam, which went online in September 2012), the station WD time series data spanning January 2005 to December 2012 were generally selected, to be consistent with the RSD time span. To convert the observed WD (in m$^3$/s) into a daily $R$ rate (in mm/day) per unit area over the MRB, the observed WD was divided by the approximate total MRB drainage area (i.e., 795,000 km$^2$). This allowed a monthly $R$ rate (in mm/month) to be calculated for the entire MRB by summing the daily $R$ rates. As mentioned above, WD is a power function of WL, so too the $R$. The specific relationship between WL and $R$ for the four selected stations, is given in Figure 2, and basic observed data—maxima, minima, means, and standard deviations—are provided in Table 1.

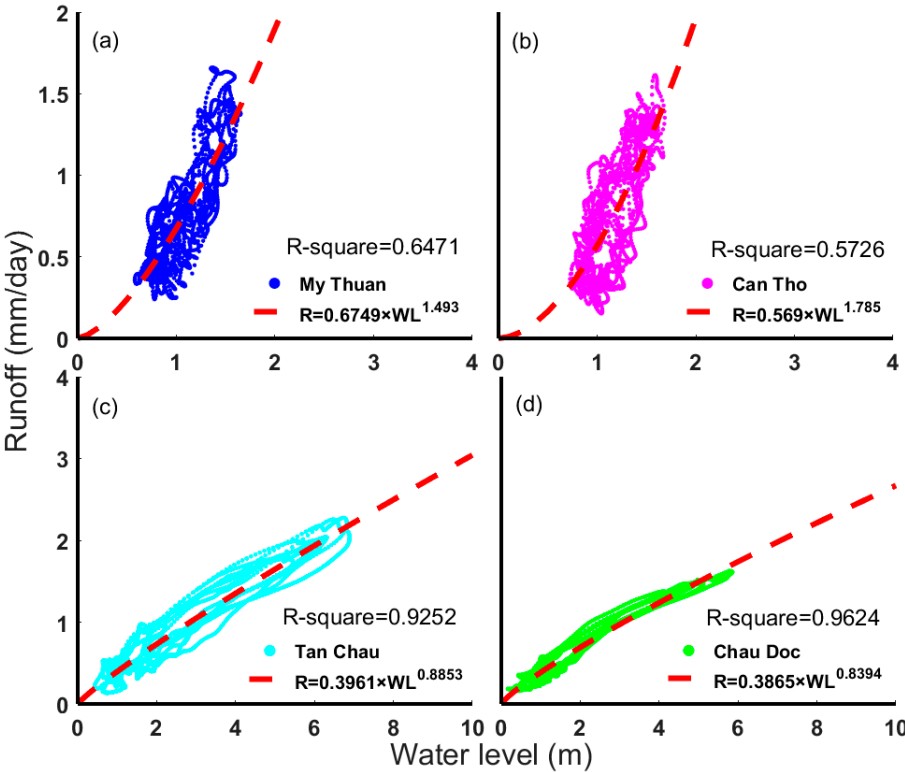

**Figure 2.** Water level and runoff rating curves for the four selected hydrological stations in the Mekong River Delta.

**Table 1.** Water discharge ($1 \times 10^4$ $m^3$/s), water level (m), and runoff (mm/month) maxima, minima, means, and standard deviations for the four selected Mekong River Delta hydrological stations.

| Variable | Station | Maximum | Minimum | Mean | Standard Deviation |
|---|---|---|---|---|---|
| Water Discharge ($1 \times 10^4$ $m^3$/s) | My Thuan | 1.854 | 0.069 | 0.661 | 0.449 |
| | Can Tho | 1.950 | 0.081 | 0.692 | 0.446 |
| | Tan Chau | 3.362 | 0.661 | 1.528 | 0.696 |
| | Chau Doc | 3.178 | 0.683 | 1.375 | 0.568 |
| Water Level (m) | My Thuan | 7.613 | 1.770 | 4.522 | 1.509 |
| | Can Tho | 4.380 | 1.714 | 3.072 | 0.655 |
| | Tan Chau | 8.481 | 0.501 | 3.688 | 1.785 |
| | Chau Doc | 7.802 | 0.244 | 2.923 | 1.449 |
| Runoff (mm/month) | My Thuan | 60.5 | 2.2 | 21.6 | 14.6 |
| | Can Tho | 63.6 | 2.6 | 22.5 | 14.6 |
| | Tan Chau | 109.6 | 21.6 | 49.8 | 22.7 |
| | Chau Doc | 103.6 | 22.3 | 44.8 | 18.5 |

Regardless of which station pair (My Thuan and Can Tho or Tan Chau and Chau Doc) were studied, the pairs' time series results were very similar and shared the same periodicities, despite differences in *R* peaks and troughs (Figure 3). Note that data were missing for the My Thuan and Can Tho station pair from January until December 2008, and so, to be consistent, the entire data of year 2008 for the station pair of Tan Chau and Chau Doc were not used.

Consistent with the descriptions in Section 2, the yearly peak and trough appear in September and April, respectively. Anomalies from February to June 2007 could be noticed for the station pair of My Thuan and Can Tho, and we speculated that the backwater effect due to ocean tides might have been intensified at this time due to local drought. No such anomaly was observed for the station pair of Tan Chau and Chau Doc.

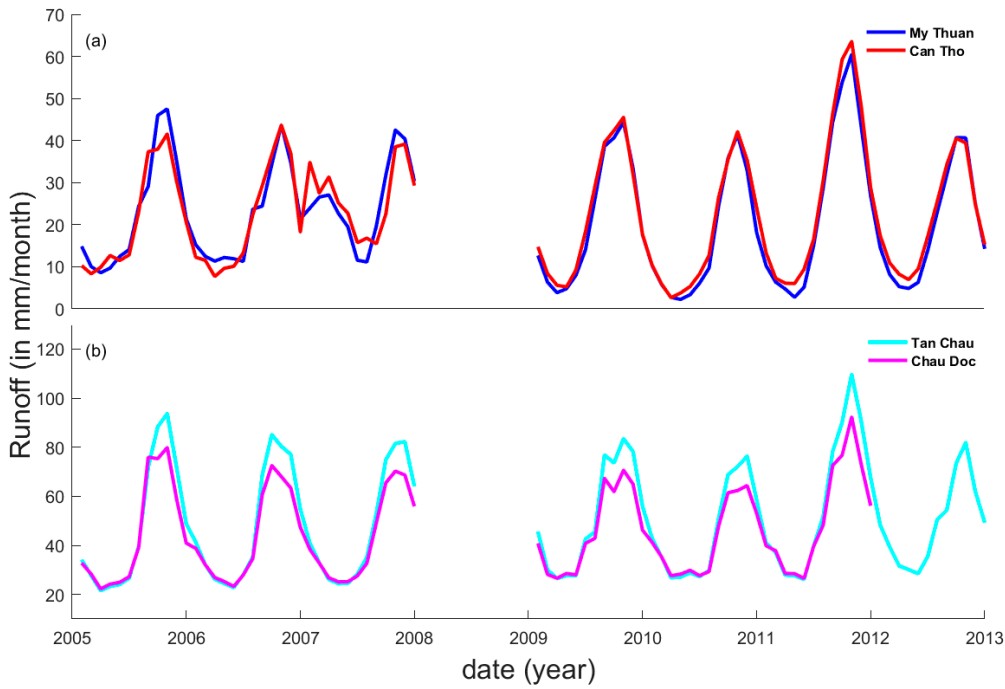

**Figure 3.** Runoff time series at (**a**) My Thuan and Can Tho stations, and (**b**) Tan Chau and Chau Doc stations.

Traditional MODIS remotely-sensed data (RSD) products (i.e., NDVI from MOD13C2, and LST from MOD11C3 data products) are accessible from the Land Processes Distribution Active Archive Center (LP DAAC) under the auspices of the NASA Earth Science Data and Information System (ESDIS) (https://lpdaac.usgs.gov/dataset_discovery/modis/modis_products_table). Both data sets were used for baseline information, to compare against *R* reconstructions.

*3.2. Remotely-Sensed Hydrological Variables and Their Standardization*

Based on the water balance concept, four kinds of hydrological variables—precipitation, evapotranspiration, water storage, and surface runoff—are described in Section 1. Accessible hydrological variable data resources are described in this section. All data, except as specified from the above and missing data, covered a time window spanning from January 2005 to December 2012, with information available at monthly sampling intervals.

3.2.1. Remotely-Sensed Precipitation and Its Standardized Index, from TRMM

TRMM is a meteorological satellite mission exclusively responsible for quantifying spatio-temporal rainfall variation in the tropical and the subtropical regions [83]. Monthly global precipitation data were derived from TRMM Rainfall Estimate L3 (TRMM_3B43 V7), with a spatial resolution of 0.25°. These data sets are available from the Goddard Earth Sciences Data and Information Services Center (GES DISC) managed by NASA (https://mirador.gsfc.nasa.gov/).

The Standardized Precipitation Index (SPI) was developed by McKee et al. [84] and has a better ability to represent precipitation deficits at different time scales. Since precipitation is temporally discrete in any one region (and does not occur every day), SPI requires a special calculation based on the standardization of cumulative probability distribution [85] (equations have not been shown since this is a well-known index).

Hence, SPI was calculated for the period 1998–2012, using the TRMM monthly precipitation data product, while extracting the calculated values between 2005 and 2012. Though a 30-year continuous precipitation time series would be preferable for climatic study, the current TRMM data time span was considered sufficient for hydrological study.

### 3.2.2. Remotely-Sensed Terrestrial Water Storage (TWS) and Its Standardized Index, from GRACE

GRACE, a joint project of NASA and Deutsches Zentrumfür Luft- und Raumfahrt (DLR), is a satellite mission that measures time-variable changes of the Earth's gravity field, thereby enabling TWS variation monitoring at the global scale. TWS data, with the spatial resolution of 1°, were computed from the CSR GRACE Level-2 Release 05 (RL05) GSM monthly gravity fields, in the form of spherical harmonic coefficients, up to degree 60. These data are freely accessible from the Center of Space Research (CSR) managed by NASA (http://www2.csr.utexas.edu/grace/RL05.html).

Note that two post-processing steps were applied to the GRACE gravity spherical harmonic coefficients to derive the TWS data. First, the GRACE $C_{20}$ term was replaced by Satellite Laser Ranging (SLR), and degree-one terms were restored to correct for geocenter motion [86,87]. Second, a de-striping process and Gaussian filtering with a radius of 350 km were applied to reduce the spatially correlated error of TWS data at higher degrees [88,89].

The derived monthly GRACE TWS (*S*) were utilized to: (i) correlate directly with the ground-based observed *R* and (ii) calculate a newly proposed GRACE drought severity index (i.e., GRACE-SI), followed by correlating with the standardized *R*.

To characterize abrupt TWS changes and drought events, GRACE-SI is a recently-derived drought severity index in a standardized form, proposed by Zhao et al. (2017) [90]. To reduce abnormal TWS anomalies, for each month of the entire time series, TWS anomaly medians, instead of TWS anomaly means, were used in the calculation of GRACE-SI [91], which is defined as shown in Equation (1).

$$SI_{i,j} = \frac{S_{i,j} - median\left(S_j\right)}{s_j} \tag{1}$$

In Equation (1), $SI_{i,j}$ and $S_{i,j}$ represent the GRACE-SI and original GRACE TWS in year *i* and month *j*, respectively, $median\left(S_j\right)$ is the median TWS value for each month, and $s_j$ is the sampled standard deviation of TWS for month *j*.

### 3.2.3. Remotely-Sensed Evapotranspiration and Its Standardized Form, from MODIS

Evapotranspiration (ET), being a total amount of water vapor transported to the atmosphere from vegetation and the land surface [92], is available from the MODIS Global Evaporation Project (MOD16A2) dataset, managed by NASA LP DAAC (https://lpdaac.usgs.gov/dataset_discovery/modis/modis_products_table). The dataset has a spatial resolution of 0.5°, from latitude 80° N to −60° N and from longitude 0° E to 360° E.

In order to capture changes in ET better, and to be consistent with the other two standardized RSHVs (i.e., SPI and SI), ET was transformed into its standardized form (called the Standardized Evapotranspiration Index (SETI)) using a similar calculation procedure, as shown in Equation (1). Hence, SETI between 2000 and 2014 was calculated while extracting the calculated values between 2005 and 2012.

## 4. Methodology and Evaluation Metrics

### 4.1. Correlative Analysis, Data Standardization Results, and Estimation Procedures

Yunnan Province, located at the upstream MRB, was selected in this study to reconstruct and estimate *R* in the MRD. All the RSD and RSHVs for the entire Yunnan Province were spatially averaged. In addition, a two-month time lag was observed between *R* and TRMM precipitation (TRMM-*P*) and MODIS evapotranspiration (MODIS-*ET*), whereas no time lag was found between *R* and GRACE TWS (GRACE-*S*) in the MRB. To reconstruct and estimate *R* in the MRD better [74], two additional pre-processing procedures were applied on three different RSHVs. Firstly, smoothing and re-scaling procedures were applied to RSHV before the correlative analysis was conducted, and secondly, a two-month time-lag shift was applied to TRMM-*P* and MODIS-*ET* to improve the estimated

performance. This form of time lag analysis has been widely used to improve hydrological modeling (Figure 4) [93].

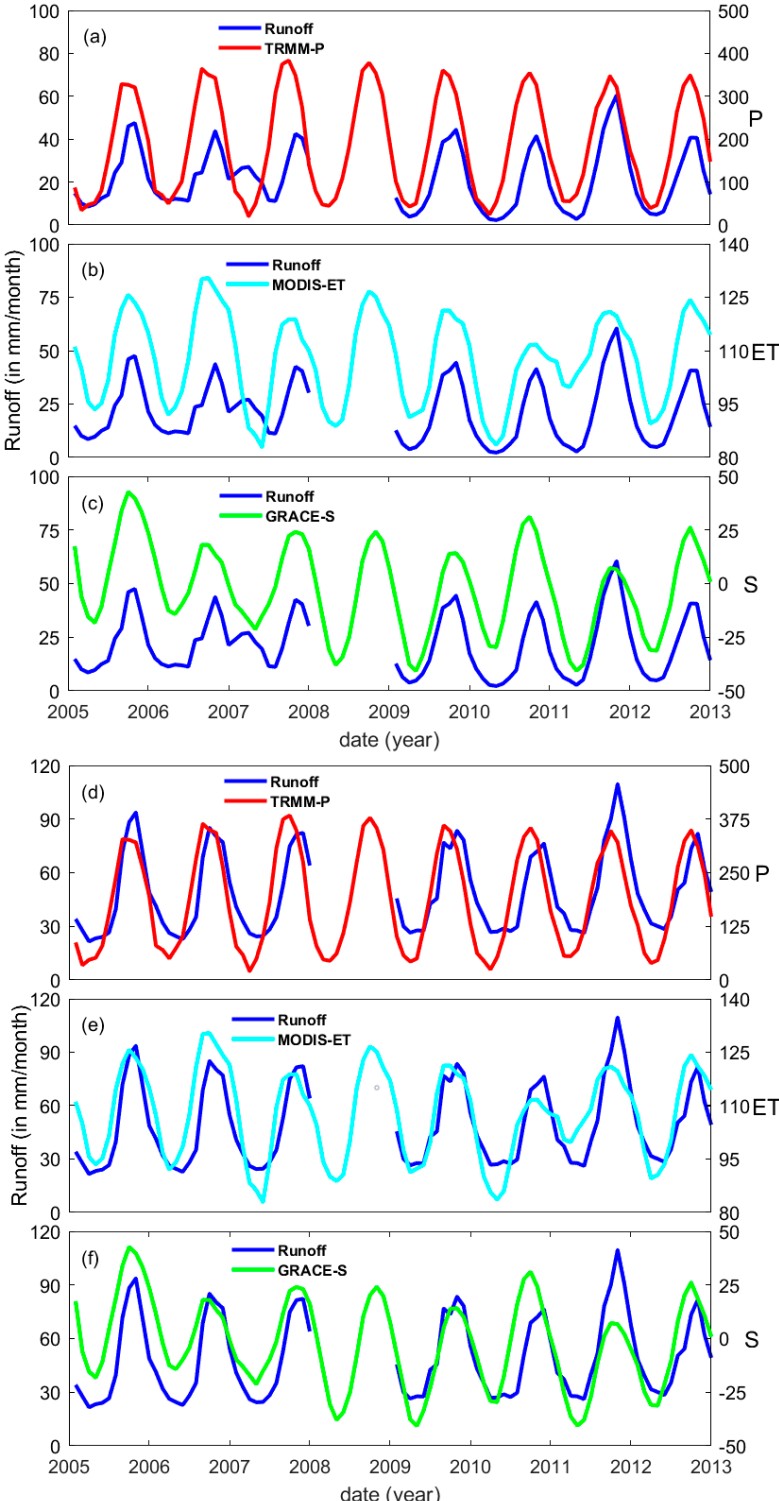

**Figure 4.** The time series of runoff against remotely-sensed hydrological variables. At My Thuan: (**a**) Tropical Rainfall Measuring Mission precipitation (TRMM-P), (**b**) Moderate Resolution Imaging Spectrometer evapotranspiration (MODIS-ET), and (**c**) Gravity Recovery and Climate Experiment terrestrial water storage (GRACE-S) values. At Tan Chau: (**d**) TRMM-P, (**e**) MODIS-ET, and (**f**) GRACE-S values.

Since the RSD (i.e., NDVI and LST) and RSHV time series were shown to have a clear correlative relationship with the observed *R* at the selected stations (i.e., My Thuan and Tan Chau stations), linear regression was applied to determine the quantitative relationship between the RSD (RSHV) and the observed *R* during the overlapping period. The *R* reconstruction formula (when $R_h = 1$) can be given as shown in Equation (2).

$$R_{i,j} = R_h \times \left( a + b \times D_{i,j} \right) \tag{2}$$

In Equation (2), $R_{i,j}$ and $D_{i,j}$ represent the reconstructed *R* and the RSD (RSHV) in year *i* and month *j*, respectively, and *a* and *b* are the offset and the slope obtained from regression fitting, respectively. $R_h$ is the ratio owing to the height difference between the two stations used for estimation (when $R_h \neq 1$), which can be calculated from the ETOPO1 1-arc-min global relief topography model [94].

These reconstruction procedures served as an internal performance assessment, whereas the same procedure applied to estimate *R* at stations other than those used for reconstruction in the MRD was referred to as an external performance assessment of the method used in this study.

To detect hydrology changes and represent regional characteristics better, the RSHVs were transformed into their standardized forms (Section 3.2) [74]. Instead of employing the sophisticated standardized runoff index (SRI) [58] that has been currently applied to renewable surface freshwater availability for agricultural products [5], the same standardization procedure was applied to *R*, in order to be comparable with the standardized RSHVs, for simplicity. *R* standardization can be achieved by using the following equation:

$$sR_{i,j} = \frac{R_{i,j} - median\left(R_j\right)}{s_j} \tag{3}$$

In Equation (3), $sR_{i,j}$ and $R_{i,j}$ represent the standardized and the observed runoff in year *i* and month *j*, respectively, $median\left(R_j\right)$ is the median runoff value for each month (Figure 5), and $s_j$ is the sampled runoff standard deviation for each separate month *j*.

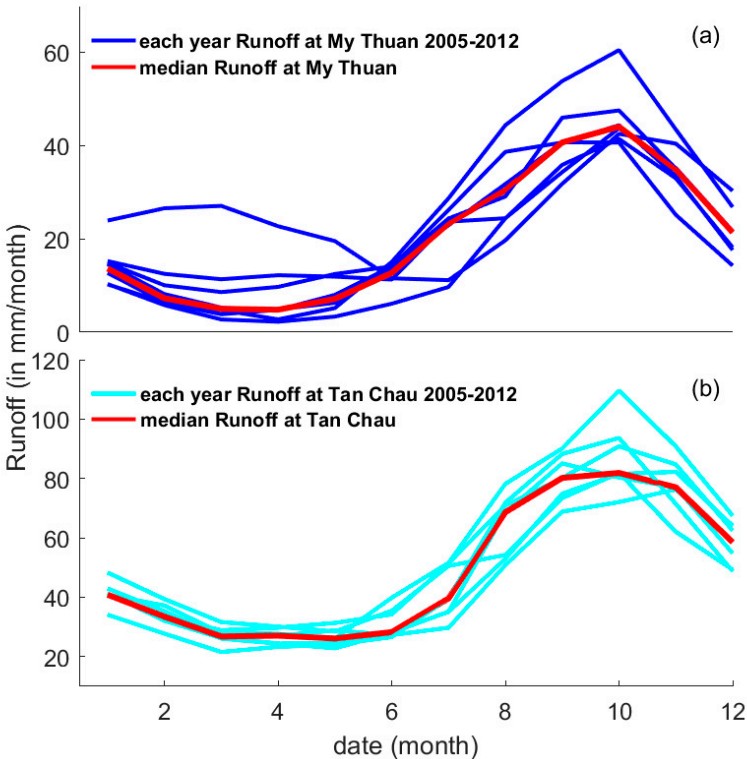

**Figure 5.** Monthly runoff, from 2005 to 2012, at (**a**) My Thuan, and (**b**) Tan Chau stations.

High correlations between the standardized RSHVs and the standardized *R* are shown in Figures 6 and 7.

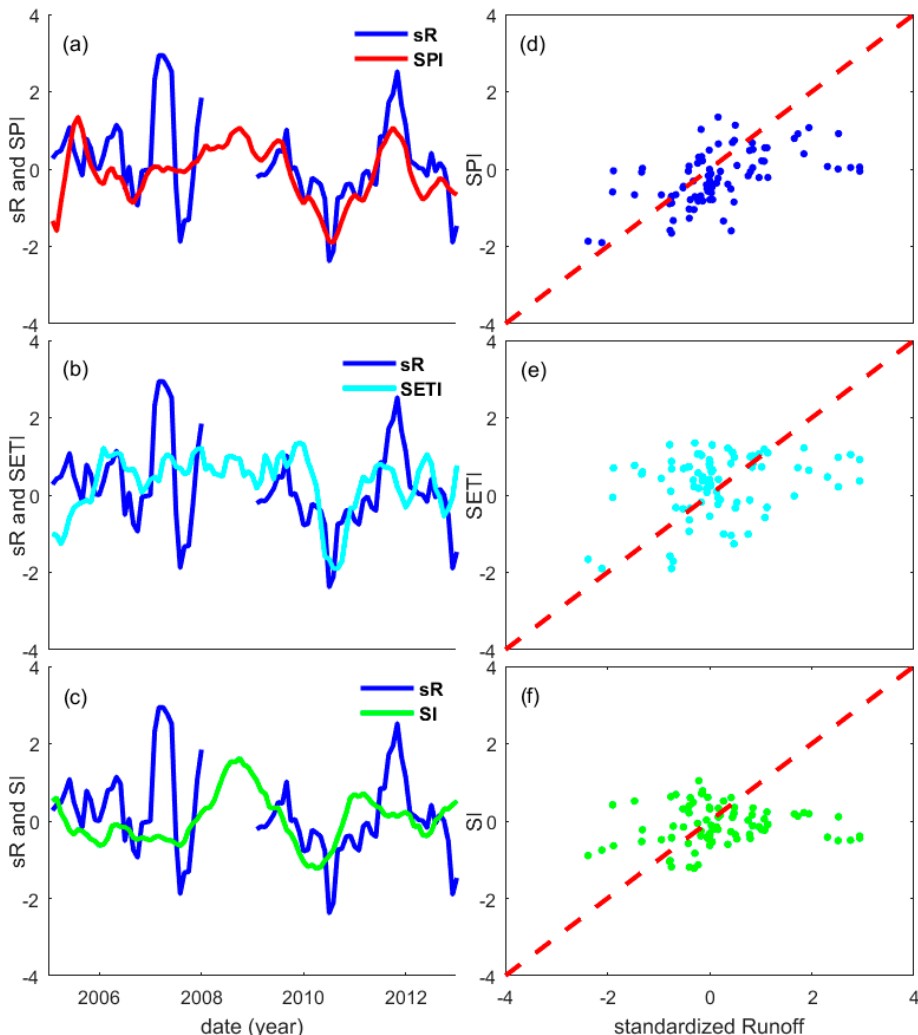

**Figure 6.** Runoff time series in the standardized form, plotted against standardized, remotely-sensed hydrological variables (RSHVs), at My Thuan: (**a**,**d**) Standardized Precipitation Index (SPI), (**b**,**e**) Standardized Evapotranspiration Index (SETI), and (**c**,**f**) severity index (SI) values. The two columns represent the time series of (left) the standardized runoff and the standardized RSHV, and (right) their scatter plots.

The water balance representation of *R* (i.e., WBR) and its standardization (SWBR) were calculated from the three RSHVs (i.e., *P*, *ET* and *S*) and their standardized forms (i.e., *SPI*, *SETI*, and *SI*), respectively. The WBR was calculated as:

$$cWBR_{i,j} = P_{i,j} - ET_{i,j} - \Delta S_{i,j} \tag{4}$$

$$WBR_{i,j} = \frac{cWBR_{i,j} - median(cWBR_j)}{s_j} \tag{5}$$

In Equations (4) and (5), $P_{i,j}$ and $ET_{i,j}$ represent *P* and *ET* in year *i* and month *j*, respectively. $\Delta S_{i,j}$ is the difference between GRACE-S in year *i*, month *j* + 1 and year *i*, month *j*. Similarly, *SWBR* can be derived as shown in (6), where $SPI_{i,j}$ and $SETI_{i,j}$ represent *SPI* and *SETI* in year *i* and month *j*, respectively. $\Delta SI_{i,j}$ is the difference between GRACE-SI in year *i*, month *j* + 1 and year *i*, month *j*.

$$SWBR_{i,j} = SPI_{i,j} - SETI_{i,j} - \Delta SI_{i,j} \tag{6}$$

A correlating procedure between the standardized *R* and all standardized forms (e.g., standardized RSHVs, WBR, and SWBR) was applied before reconstruction and estimation. Then, the reconstructed ($R_h = 1$) and estimated ($R_h \neq 1$ from values listed in Tables 1 and 2) *R*, derived from the standardized RSHVs and their water balance representations (i.e., WBR and SWBR), were defined, as shown in Equation (7).

$$R_{i,j} = R_h \times \left( I_{i,j} \times s_j + median\left(R_j\right) \right) \tag{7}$$

In Equation (7), $R_{i,j}$ and $I_{i,j}$ represent the reconstructed (estimated) *R* and the standardized hydrological variables (i.e., *SPI*, *SETI* and *SI*), and their water balance representations (i.e., WBR and SWBR), in year *i* and month *j*, respectively.

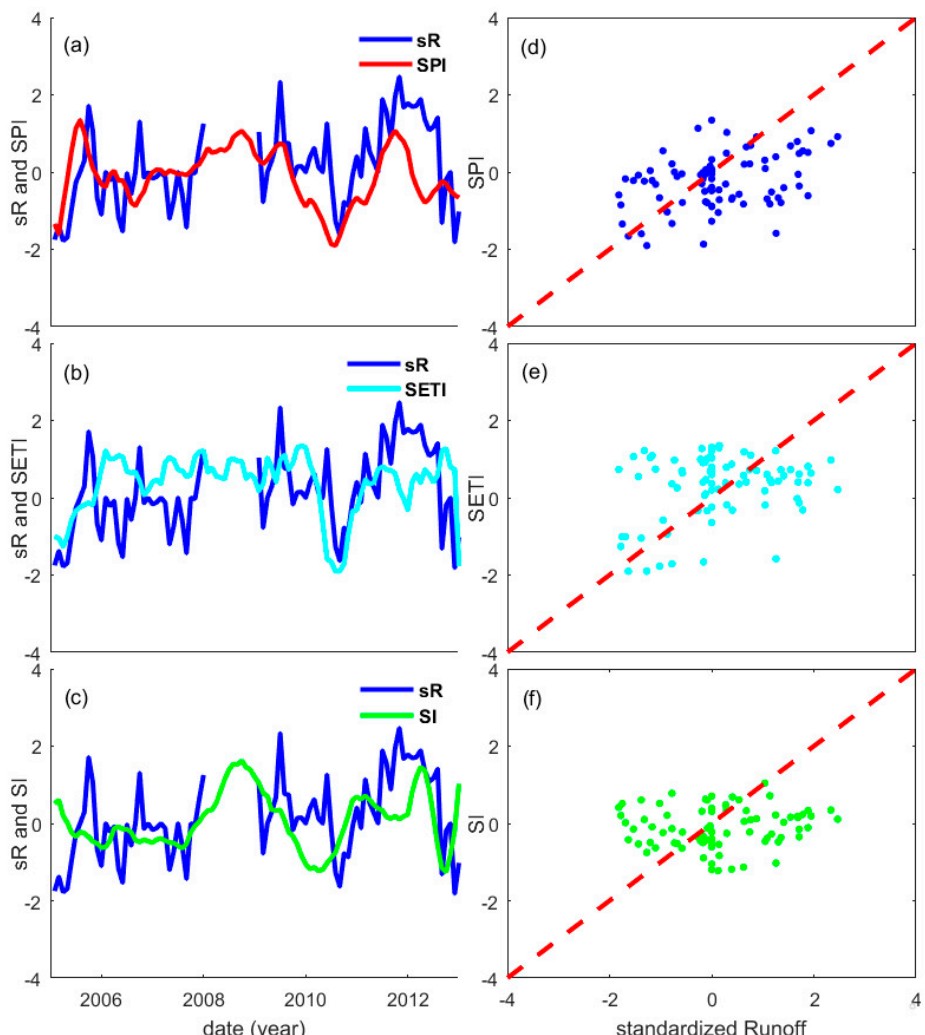

**Figure 7.** Runoff time series in the standardized form plotted against the standardized, remotely-sensed hydrological variables (RSHVs) at Tan Chau station: (**a**,**d**) SPI, (**b**,**e**) SETI, and (**c**,**f**) SI values. The two columns represent the time series of (left) the standardized runoff and the standardized RSHV and (right) their scatter plots.

### 4.2. Result Evaluation Metrics

To conduct a performance evaluation for the reconstructed *R* against the observed *R* at in situ stations in the MRD, the Pearson correlation coefficient (PCC), root mean square error (RMSE), and Nash–Sutcliffe efficiency model (NSE) were selected.

PCC, which represents the linear relationship between two variables, is defined as shown in Equation (8).

$$PCC = \frac{\sum_{i=1}^{N}\left(R_0^i - \overline{R_0}\right)\left(R_m^i - \overline{R_m}\right)}{\sqrt{\sum_{i=1}^{N}\left(R_0^i - \overline{R_0}\right)^2}\sqrt{\sum_{i=1}^{N}\left(R_m^i - \overline{R_m}\right)^2}} \tag{8}$$

RMSE, which is an accuracy indicator, is defined as shown in Equation (9).

$$RMSE = \sqrt{\frac{\sum_{i=1}^{N}\left(R_m^i - R_0^i\right)^2}{N}} \tag{9}$$

The NSE, which is a performance coefficient applied to evaluate the estimated $R$ against the observed $R$ [95], ranges from $-\infty$ to 1, and the closer the NSE value is to 1, the better the estimation's performance. It is calculated as shown in Equation (10), where $R_0$ and $R_m$ represent the observed and estimated $R$, respectively, and $\overline{R_0}$ is the mean of $R_0$.

$$NSE = 1 - \frac{\sum_{i=1}^{N}\left(R_m^i - R_0^i\right)^2}{\sum_{i=1}^{N}\left(R_0^i - \overline{R_0}\right)^2} \tag{10}$$

## 5. Results and Discussion

In this section, the performance of the reconstructed and the estimated $R$ time series from the traditional RSD (i.e., NDVI and LST), RSHV (i.e., *P, ET, S*), and their respective standardized forms (i.e., *SPI, SETI,* and *SI*), including water balance representations (i.e., WBR and SWBR) have been presented and evaluated, using internal and external assessment, respectively. The $R$ time series from the My Thuan and Tan Chau stations were independently chosen to reconstruct $R$ and to conduct the internal assessment. Other station time series served to validate the estimation via the external assessment. Since the My Thuan and Can Tho station pair is ~120 km closer to the estuary mouth than the Tan Chau and Chau Doc pair, their combined assessments could provide insight into the impact of the backwater effect (due to landward ocean tidal propagation) on both the reconstructed and estimated $R$.

Both NDVI and LST, which served as baseline results for RSHVs (and their standardized forms), and WBR and SWBR captured the temporal pattern of the observed $R$ from My Thuan station well (Figure 6a,b). Distinct discrepancies in peaks and troughs between the NDVI- and LST-reconstructed $R$ and the observed $R$ from My Thuan station were observed, whereas these discrepancies were reduced in the Tan Chau station time series (Figure 8c,d). The discrepancies were considered most likely to be due to the impact of the tidal backwater effect on the estimated $R$ at My Thuan station, being closer to the estuary mouth than that of Tan Chau station.

RSHVs manifested a performance in $R$ reconstruction similar to that of the traditional RSD (Table 2), while the standardized RSHVs performed better (Figure 9a–c) at replicating the peaks and the troughs because the systematic influences should be mitigated by the standardization process. A similar situation, explaining the impact of the backwater effect, applied to the Tan Chau station time series, where reduction in the discrepancies in the peaks and the troughs was also shown (Figure 9d–f, and Table 3). Tables 2 and 3 display the consistency of the results, as demonstrated by the PCC and NSE values, showing that an efficiency gain for the reconstructed $R$ in the downstream MRB was achieved through the standardization process when compared to the direct correlative analysis.

Compared with the above traditional RSD-reconstructed $R$, RSHV-reconstructed $R$, and their standardization reconstructed $R$, two WBR-reconstructed $R$s (i.e., WBR and SWBR) achieved even better reconstruction outcomes in the MRB (Figure 10). Noting that the WBRs were derived from water balance equations (i.e., Equations (5) and (6)), the systematic errors existing in the RSHV data and introduced through standardization were further reduced via the subtraction process (i.e., $S_{i+1} - S_i$ and $P - ET$). As a result, $R$ reconstructed from WBR achieved a relatively good outcome (Figure 8a,b),

while *R* reconstructed from SWBR displayed the best performance among all reconstructions in the MRB (Figure 8c,d), which could be a reflection of the ability of the standardization process to filter out remaining systematic effects further.

Using Equations (2) and (7) from Section 4.1, *R* estimation could be performed based on the constructed correlative relationship between the traditional RSD, RSHV (and its standardization), and WBR at locations other than the My Thuan or Tan Chau stations. This allowed us to assess the new approach presented in this study externally.

**Table 2.** Internal assessment of surface runoff (*R*), reconstructed using data from My Thuan station, and external assessment of *R*, estimated for Can Tho and Tan Chau stations. The legend 'My Thuan estimate Can Tho' (or 'Tan Chau') means that the *R* for Can Tho (or Tan Chau) was estimated based on the reconstructed relationships between the *R* from My Thuan station and the abovementioned variables.

| Station | Variables | | PCC | RMSE (mm) | NSE |
|---|---|---|---|---|---|
| My Thuan ($R_h = 1$) | Traditional RSD | NDVI | 0.766 | 9.1 | 0.585 |
| | | LST | 0.808 | 8.3 | 0.651 |
| | Hydrological Variables | TRMM-P | 0.773 | 9.0 | 0.603 |
| | | MODIS-ET | 0.771 | 9.1 | 0.600 |
| | | GRACE-S | 0.741 | 9.6 | 0.556 |
| | Hydrological Indices | SPI | 0.905 | 6.1 | 0.811 |
| | | SETI | 0.903 | 6.2 | 0.808 |
| | | SI | 0.897 | 6.3 | 0.799 |
| | Water-balance Representations | WBR | 0.898 | 6.3 | 0.802 |
| | | SWBR | 0.910 | 5.9 | 0.822 |
| My Thuan estimate Can Tho ($R_h = 1.05$) | Traditional RSD | NDVI | 0.722 | 9.8 | 0.507 |
| | | LST | 0.764 | 9.1 | 0.575 |
| | Hydrological Variables | TRMM-P | 0.752 | 9.5 | 0.576 |
| | | MODIS-ET | 0.741 | 9.7 | 0.562 |
| | | GRACE-S | 0.711 | 10.1 | 0.513 |
| | Hydrological Indices | SPI | 0.860 | 7.3 | 0.730 |
| | | SETI | 0.856 | 7.4 | 0.722 |
| | | SI | 0.852 | 7.4 | 0.716 |
| | Water-balance Representations | WBR | 0.854 | 7.3 | 0.725 |
| | | SWBR | 0.865 | 7.1 | 0.738 |
| My Thuan estimate Tan Chau ($R_h = 2.20$) | Traditional RSD | NDVI | 0.813 | 11.7 | 0.576 |
| | | LST | 0.867 | 13.1 | 0.668 |
| | Hydrological Variables | TRMM-P | 0.937 | 9.6 | 0.822 |
| | | MODIS-ET | 0.850 | 14.6 | 0.588 |
| | | GRACE-S | 0.772 | 15.8 | 0.518 |
| | Hydrological Indices | SPI | 0.923 | 12.2 | 0.714 |
| | | SETI | 0.928 | 12.1 | 0.720 |
| | | SI | 0.937 | 11.2 | 0.758 |
| | Water-balance Representations | WBR | 0.930 | 12.3 | 0.709 |
| | | SWBR | 0.934 | 10.7 | 0.781 |

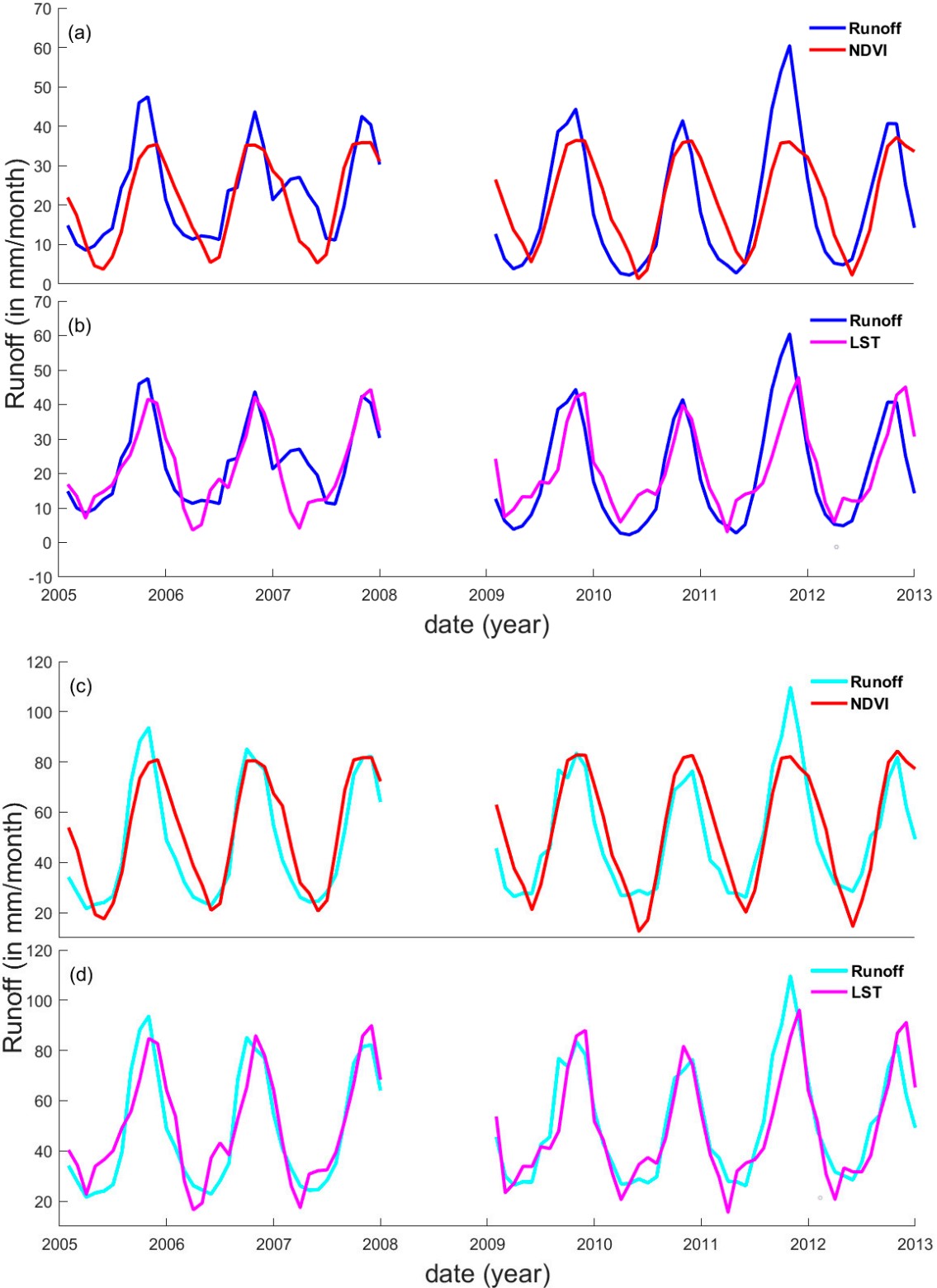

**Figure 8.** Reconstructed runoff estimates based on (**a**,**c**) Normalized Difference Vegetation Index (NDVI) and (**b**,**d**) Land Surface Temperature (LST), (**a**,**b**) My Thuan station, (**c**,**d**) Tan Chau station.

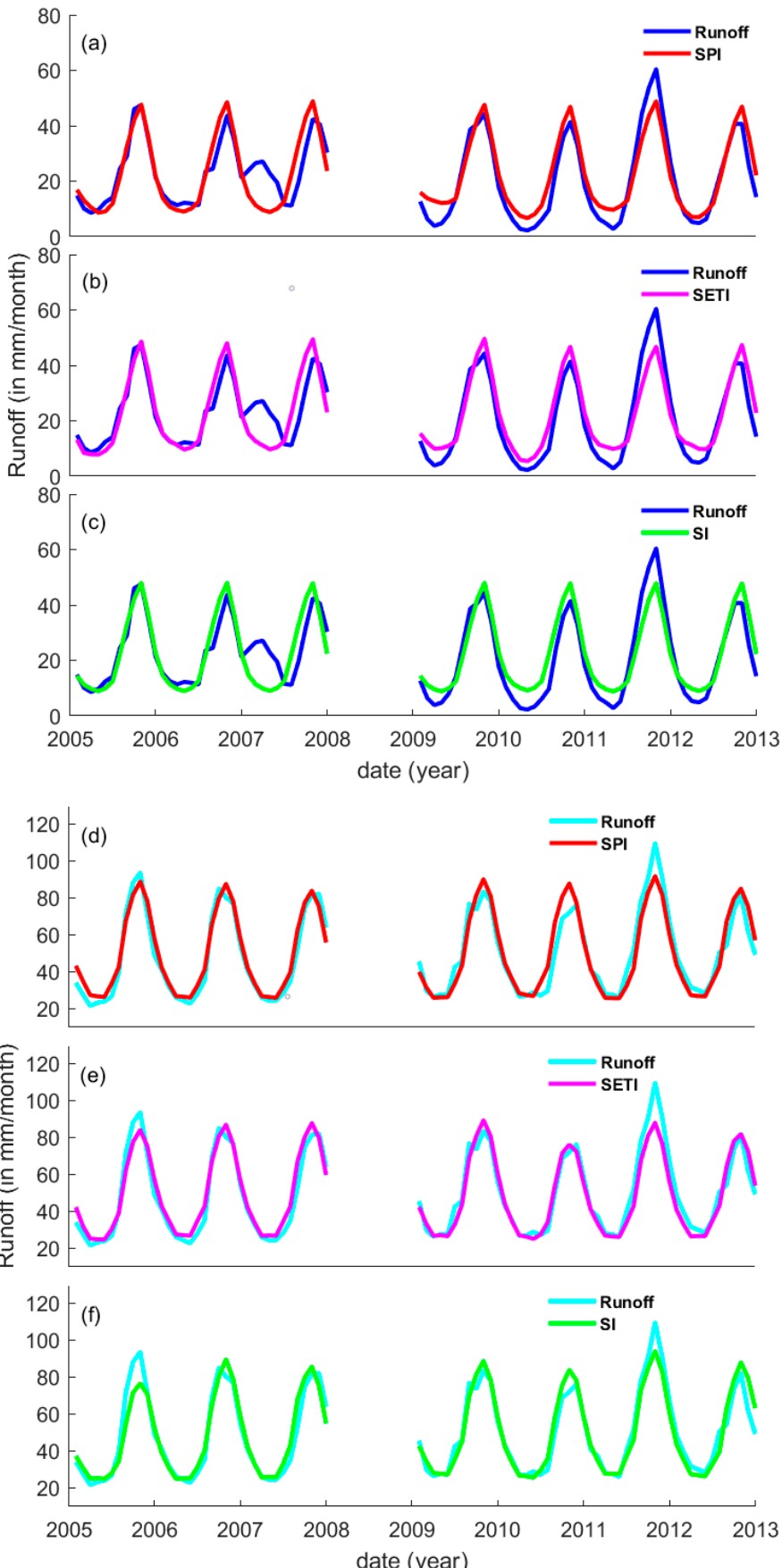

**Figure 9.** Reconstructed runoff estimates, based on (**a**,**d**) SPI, (**b**,**e**) SETI, and (**c**,**f**) SI, at (**a**–**c**) My Thuan and (**d**–**f**) Tan Chau stations.

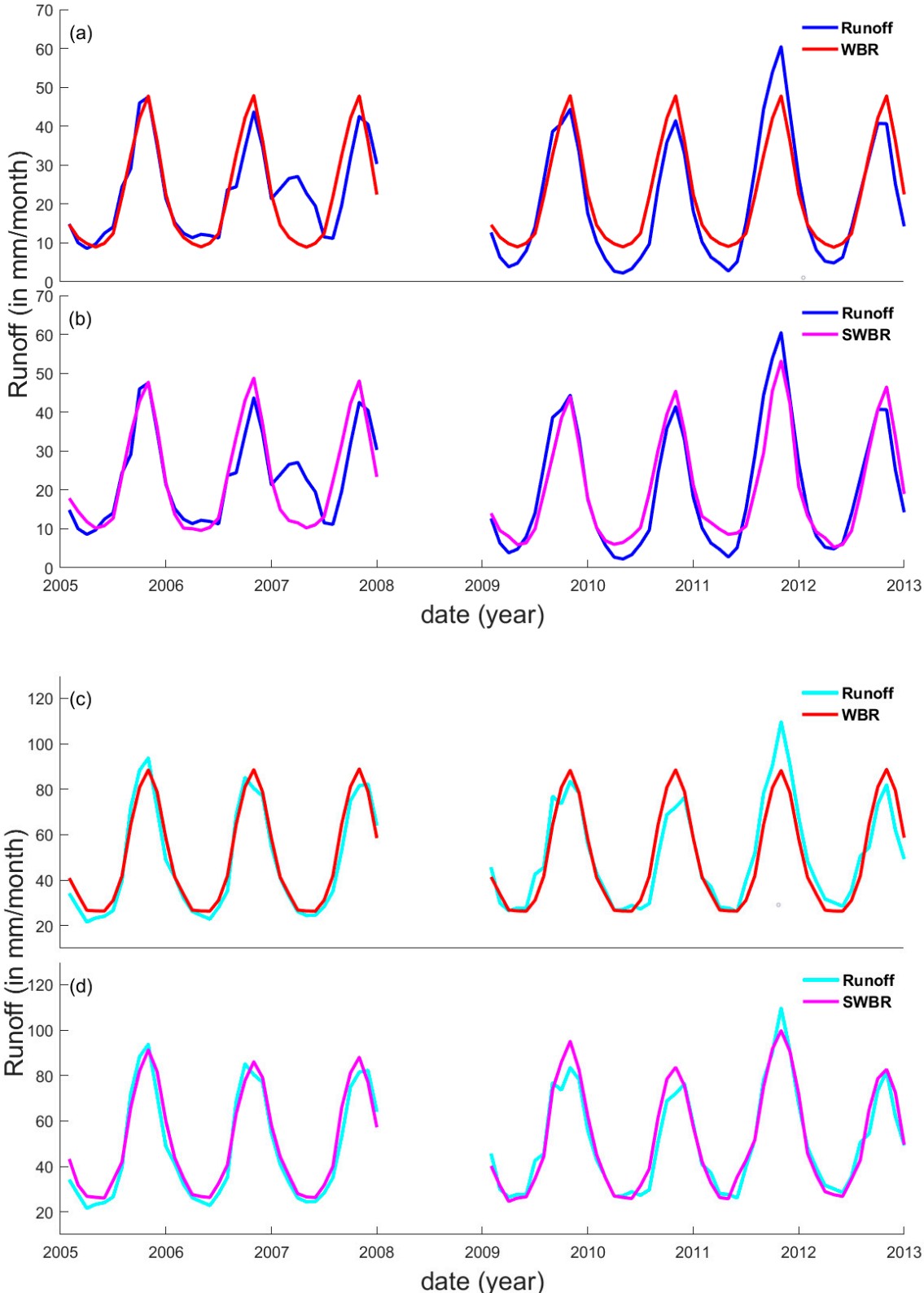

**Figure 10.** Reconstructed runoff estimates, based on (**a**,**c**) water balance equation (WBR) and (**b**,**d**) standardized water balance representation (SWBR), calculated from the WBR and its standardized form SWBR at (**a**,**b**) My Thuan and (**c**,**d**) Tan Chau stations.

**Table 3.** Internal assessment of *R*, reconstructed using data from Tan Chau station, and external assessment of the *R*, estimated for Chau Doc and My Thuan stations. The legend 'Tan Chau estimate Chau Doc' (or 'My Thuan') means that the *R* for Chau Doc (or My Thuan) was estimated based on the reconstructed relationships between the *R* from Tan Chau station and the abovementioned variables.

| Station | Variables | | PCC | RMSE (mm) | NSE |
|---|---|---|---|---|---|
| Tan Chau $(R_h = 1)$ | Traditional RSD | NDVI | 0.863 | 11.5 | 0.745 |
| | | LST | 0.866 | 11.4 | 0.750 |
| | Hydrological Variables | TRMM-P | 0.935 | 8.1 | 0.874 |
| | | MODIS-ET | 0.904 | 10.7 | 0.818 |
| | | GRACE-S | 0.768 | 14.6 | 0.590 |
| | Hydrological Indices | SPI | 0.962 | 6.4 | 0.921 |
| | | SETI | 0.960 | 6.5 | 0.917 |
| | | SI | 0.952 | 7.0 | 0.905 |
| | Water-balance Representations | WBR | 0.955 | 6.8 | 0.912 |
| | | SWBR | 0.965 | 5.9 | 0.931 |
| Tan Chau estimate Chau Doc $(R_h = 0.76)$ | Traditional RSD | NDVI | 0.791 | 11.8 | 0.591 |
| | | LST | 0.757 | 12.7 | 0.525 |
| | Hydrological Variables | TRMM-P | 0.858 | 10.8 | 0.699 |
| | | MODIS-ET | 0.816 | 11.2 | 0.626 |
| | | GRACE-S | 0.766 | 11.9 | 0.576 |
| | Hydrological Indices | SPI | 0.887 | 9.2 | 0.749 |
| | | SETI | 0.890 | 9.0 | 0.760 |
| | | SI | 0.865 | 10.0 | 0.703 |
| | Water-balance Representations | WBR | 0.883 | 9.4 | 0.739 |
| | | SWBR | 0.900 | 8.5 | 0.785 |
| Tan Chau estimate My Thuan $(R_h = 0.45)$ | Traditional RSD | NDVI | 0.766 | 9.3 | 0.564 |
| | | LST | 0.808 | 8.7 | 0.617 |
| | Hydrological Variables | TRMM-P | 0.808 | 8.6 | 0.633 |
| | | MODIS-ET | 0.821 | 8.4 | 0.644 |
| | | GRACE-S | 0.703 | 10.3 | 0.470 |
| | Hydrological Indices | SPI | 0.864 | 7.5 | 0.716 |
| | | SETI | 0.880 | 7.3 | 0.735 |
| | | SI | 0.883 | 7.2 | 0.740 |
| | Water-balance Representations | WBR | 0.864 | 7.6 | 0.714 |
| | | SWBR | 0.881 | 7.1 | 0.750 |

In general, all the estimated *R* time series were slightly less accurate than those recorded for My Thuan (Table 2) and Tan Chau (Table 3). This could be partly attributable to error propagation in the reverse process. However, the relative ranking of their performances also remained the same, no matter whether reconstructed with either My Thuan or Tan Chau data.

By assessing the differences in the result evaluation metrics between Tables 2 and 3, it became clear that the tidal backwater effect was another error source. It accounted for ~5% increase in PCC, and 4–5 mm changes in RMSE for the standardized RSHV, and WBR and SWBR, no matter whether Tan Chau (My Thuan) or My Thuan (Tan Chau) estimates were used (Table 4). Similar evaluation metrics were confirmed in the differences between Tau Chau and My Thuan reconstructed *R* (Table 4). As a result, when the 4–5 mm changes in RMSE were normalized in the runoff peak-to-peak range (i.e., ~45 mm), the backwater effect on the estimated *R* accounted for 9–11% of the error when tidally dominated hydrological stations were selected.

This indicated that careful station selection was essential. Overall, our presented approach, based on standardization, proved itself better than direct correlative analysis, making it a plausible method to estimate *R* in the MRD at an ungauged station, using data from a gauged location.

**Table 4.** Differences in the result evaluation metrics between Tables 2 and 3. Pearson correlation coefficient (PCC) and Nash–Sutcliffe Model efficiency coefficient (NSE) values shown as percentages.

| Station | Variables | | ΔPCC | ΔRMSE (mm) | ΔNSE |
|---|---|---|---|---|---|
| Tan Chau minus My Thuan | Traditional RSD | NDVI | 9.7% | 2.4 | 16.0% |
| | | LST | 5.8% | 3.1 | 9.9% |
| | Hydrological Variables | TRMM-P | 16.2% | −0.9 | 27.1% |
| | | MODIS-ET | 13.3% | 1.6 | 21.8% |
| | | GRACE-S | 2.7% | 5.0 | 3.4% |
| | Hydrological Indices | SPI | 5.7% | 0.3 | 11.0% |
| | | SETI | 5.7% | 0.3 | 10.9% |
| | | SI | 5.5% | 0.7 | 10.6% |
| | Water-balance Representations | WBR | 5.7% | 0.5 | 11.0% |
| | | SWBR | 5.5% | 0.0 | 10.9% |
| Tan Chau estimate Chau Doc minus My Thuan estimate Can Tho | Traditional RSD | NDVI | 6.9% | 2.0 | 8.4% |
| | | LST | −0.7% | 3.6 | −5.0% |
| | Hydrological Variables | TRMM-P | 10.6% | 1.3 | 12.3% |
| | | MODIS-ET | 7.5% | 1.5 | 6.4% |
| | | GRACE-S | 5.5% | 1.8 | 6.3% |
| | Hydrological Indices | SPI | 2.7% | 1.9 | 1.9% |
| | | SETI | 3.4% | 1.6 | 3.8% |
| | | SI | 1.3% | 2.6 | −1.3% |
| | Water-balance Representations | WBR | 2.9% | 2.1 | 1.4% |
| | | SWBR | 3.5% | 1.4 | 4.7% |
| Tan Chau estimate My Thuan minus My Thuan estimate Tan Chau | Traditional RSD | NDVI | −4.7% | −2.4 | −1.2% |
| | | LST | −5.9% | −4.4 | −5.1% |
| | Hydrological Variables | TRMM-P | −12.9% | −1.0 | −18.9% |
| | | MODIS-ET | −2.9% | −6.2 | 5.6% |
| | | GRACE-S | −6.9% | −5.5 | −4.8% |
| | Hydrological Indices | SPI | −5.9% | −4.7 | 0.2% |
| | | SETI | −4.8% | −4.8 | 1.5% |
| | | SI | −5.4% | −4.0 | −1.8% |
| | Water-balance Representations | WBR | −6.6% | −4.7 | 0.5% |
| | | SWBR | −5.3% | −3.6 | −3.1% |

## 6. Conclusions

Contrary to the traditional practice of using indirect data from remote sensing (i.e., RSD) for surface runoff (*R*) reconstruction, remotely-sensed hydrological variables (RSHVs), which represent more direct causal relationships, were investigated at a monthly temporal scale. Due to the geographic characteristics of the Mekong River Basin (MRB), a two-month time lag shift was applied between the observed *R*, the TRMM precipitation, and MODIS evapotranspiration data. It was found that the standardized, spatially-averaged RSHVs from upstream were better for reconstructing *R* than the traditional RSD from the downstream MRB. Internal assessments also showed that the standardized RSHVs attained PCC and NSE values of 0.91 (0.96) and 0.81 (0.92) for My Thuan and Tan Chau stations, respectively.

A new approach, based on water balance representation and its standardization (i.e., WBR and SWBR) has been proposed for further accuracy improvement. Using reduction of systematic error among RSHVs by the subtraction process, results from WBR and SWBR were similar and gave better estimation performances when compared to both RSD and RSHV. SWBR displayed the best reconstructed ability, with a PCC of 0.91 (0.97), an RMSE of 5.9 (5.9) mm, and an NSE of 0.84 (0.93) for My Thuan (Tan Chau stations), whereas traditional RSD reconstructed *R* showed less accurate

results (PCC of 0.81 (0.87), RMSE of 8.5 (11.4) mm, and NSE of 0.66 (0.75) for LST). Comparing results of reconstructed and estimated $R$s between My Thuan and Tan Chau revealed that the tidally-induced backwater effect on the estimated $R$ accounted for 9–11% of the error source. Meanwhile, similar performances were given for the estimated $R$, as validated externally through unused, ground-based measurements from other station locations.

We anticipate that the presented standardized water balance representation (SWBR) can be applied to basin-wide discharge estimation without assistance from in situ, ground-based, $R$ time series measurements. Considering the fact that different basins around the globe exhibit different hydrographic and hydro-climatic characteristics, further experiments with different river basins should be conducted in the future to confirm the feasibility of the presented approach.

In addition, owing to the ongoing improvement in the retrieval algorithms for satellite measurements, satellite-based RSHV data with higher temporal resolution—such as daily TRMM precipitation [96], 8-day MODIS evapotranspiration [97], and daily GRACE terrestrial water storage data products [98]—is increasingly available. The research described herein can be extended to include these improved data products into the present work.

**Author Contributions:** L.Z. performed data post-processing and wrote the manuscript. H.S.F. designed an initial concept and experiment, collected data, performed data pre-processing, obtained funding, interpreted the results, and revised the manuscript. Z.M. performed data pre-processing and also interpreted the results. Q.C. performed GRACE data pre-processing and contributed to the revised manuscript.

**Funding:** This research was funded by the National Natural Science Foundation of China (NSFC), grants number 41674007, and 41429401.

**Acknowledgments:** The authors appreciate the river water discharge data obtained from the Mekong River Commission (MRC), purchased using NSFC Grant No.: 41374010.

**Conflicts of Interest:** The authors declare no conflict of interest.

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
