# Peer review of "Upstream Remotely-Sensed Hydrological Variables and Their Standardization for Surface Runoff Reconstruction and Estimation of the Entire Mekong River Basin"

_remotesensing, doi:10.3390/rs11091064_

Round 1

Reviewer 1 Report

I believe that ms. is now ready for publication.

Author Response

Thank you.

Reviewer 2 Report

The manuscript entitled ”Upstream Remotely-Sensed Hydrological Variables and their Standardization for Downstream Water Discharge Reconstruction and Prediction in the Mekong River Basin” by Zhou et al. submitted to Remote Sensing. I suggest the manuscript should be accepted for publication after minor edits:

- Consider removing “estuarine” from the title. At the last line of the abstract, the authors stated, “Comparison between hydrological stations at the Mekong River Delta entrance and near estuary mouth”.

- “The Mekong River Delta (MRD), close to the estuary mouth where there is …”: there are several river mouths; please revise the whole manuscript accordingly.

- Consider removing “Kratie and Stung Treng stations were considered to be too far away from the estuary mouth, and were also not selected”.

- Consider revising the font size and resolution of Figures 1 & 2.

- Consider choosing better keywords.

Author Response

The manuscript entitled ”Upstream Remotely-Sensed Hydrological Variables and their Standardization for Downstream Water Discharge Reconstruction and Prediction in the Mekong River Basin” by Zhou et al. submitted to Remote Sensing. I suggest the manuscript should be accepted for publication after minor edits:

- Consider removing “estuarine” from the title. At the last line of the abstract, the authors stated, “Comparison between hydrological stations at the Mekong River Delta entrance and near estuary mouth”.

Response: Yes, we removed it from the title.

- “The Mekong River Delta (MRD), close to the estuary mouth where there is …”: there are several river mouths; please revise the whole manuscript accordingly.

Response: Yes, we did revise the whole manuscript according to your above instruction.

- Consider removing “Kratie and Stung Treng stations were considered to be too far away from the estuary mouth, and were also not selected”.

Response: Yes, we removed this sentence.

- Consider revising the font size and resolution of Figures 1 & 2.

Response: Yes, we enlarged the font size, finer axes and resolution, etc.

- Consider choosing better keywords.

Response: Yes, we revised the keywords to "Runoff; Water balance standardization; GRACE satellite gravimetry; remote sensing hydrology; Mekong River Basin".

Reviewer 3 Report

I am glad to see that the authors accounted for my suggestions

as a minor comment, I would reduce the number acronyms in the abstract (to be checked according to the journal style recommendations)

Author Response

I am glad to see that the authors accounted for my suggestions

Response: Thank you. We did our best to revise it accordingly.

as a minor comment, I would reduce the number acronyms in the abstract (to be checked according to the journal style recommendations)

Response: In case reducing the number of acronyms, it would lengthen the abstract significantly. This is our concern. We would let the administrative editor to decide and help on that (as stated from your above instruction). 

Reviewer 4 Report

I am happy that the authors handle well my reviews. I think this revised manuscript will be worthy of publishing on Remote Sensing Journal after some minor corrections as below.

Line 208 … the total MRB drainage area (i.e., 795, 000 km2) … --> the approximately total RMB drainage area (i.e., 795, 000 km2). The reason for adding “approximately” term is that 795,000 km2 is a total area of RMB, considering the river outlet at the estuary mouth. Since selected four hydrological used in this study are several hundred kilometers away from the estuary mouth, their basin areas are smaller than 795,000 km2. However, considering the differences are small, we can use the total MRB drainage area to represent the basin area for each station.

Figure 2. The study calculated Runoff from Water Level, so it is better to present Y-axis as Runoff and X-axis as Water Level. Please add R-squared for each rating curve.

Figure 4. Double check the second-Y axis.

Author Response

I am happy that the authors handle well my reviews. I think this revised manuscript will be worthy of publishing on Remote Sensing Journal after some minor corrections as below.

Response: Thank you. We did our best to revise it accordingly.

Line 208 … the total MRB drainage area (i.e., 795, 000 km2) … --> the approximately total RMB drainage area (i.e., 795, 000 km2). The reason for adding “approximately” term is that 795,000 km2 is a total area of RMB, considering the river outlet at the estuary mouth. Since selected four hydrological used in this study are several hundred kilometers away from the estuary mouth, their basin areas are smaller than 795,000 km2. However, considering the differences are small, we can use the total MRB drainage area to represent the basin area for each station.

Response: Yes, you are correct. We add "approximately" to this sentence accordingly.

Figure 2. The study calculated Runoff from Water Level, so it is better to present Y-axis as Runoff and X-axis as Water Level. Please add R-squared for each rating curve.

Response: Yes, we revised the figure accordingly.

Figure 4. Double check the second-Y axis.

Response: Thank you for your reminder and taking great care of details. We revised it accordingly. 

This manuscript is a resubmission of an earlier submission. The following is a list of the peer review reports and author responses from that submission.

Round 1

Reviewer 1 Report

Authors propose inovative method for a river discharge reconstruction using remote sensing data.

I have only some minor remarks and suggestions: 

l50: Can we really predict WD? also in l122

l126: write: "longest in the world"

l210 write: "Note that three'

Authors should explain or discuss in detail the existence of relatively large differences between investigated quantities presented in Figs. 6-8 over a first six months of 2007!  

Author Response

Authors propose inovative method for a river discharge reconstruction using remote sensing data.

Response: Thank you.

I have only some minor remarks and suggestions: 

l50: Can we really predict WD? also in l122

Response: No. You are right. We changed to "estimation or estimated" for the entire manuscript, since we were estimating the WD time series at ungauged locations (in space).

l126: write: "longest in the world"

Response: Done.

l210 write: "Note that three'

Response: Done.

Authors should explain or discuss in detail the existence of relatively large differences between investigated quantities presented in Figs. 6-8 over a first six months of 2007!  

Response: Actually, it is also presented in the in situ ground data in Figure 2. Our explanation is that Can Tho and My Thuan stations are too close to the estuary mouth (~100 km), subject to the effect of backwater on the observed discharge due to landward ocean tidal propagation into the estuary channel. The dry season even make lesser discharge, in addition to dam operation, etc.

Since another reviewer Can Tho and My Thuan stations are too close to the estuary mouth (~100 km) that are influenced by ocean tides, another stations pair (i.e. Tan Chau and Chau Doc which are 220 km away from the estuary mouth) were further employed and tested for the study. We found that these large differences disappeared, implying that it is due to the backwater effect.

Reviewer 2 Report

Please find in attached!

Author Response

The manuscript entitled “: Upstream Remotely-Sensed Hydrological Variables and their Standardization for Downstream Water Discharge Reconstruction and Prediction in the Mekong River Basin” proposed a downstream discharge prediction from upstream remote-based hydrological variables at Mekong River Basin. The manuscript is interesting in a novel method. However, the authors need to design again their case study to have scientific soundness merit. I am confused with the technical term River Discharge and Runoff using in the paper. If authors want to predict River Discharge (or Runoff ?), I suggest they need to select another gauge station. Both My Thuan and Can Tho hydrological stations are in a tidal affected area in Mekong River where Discharge can be negative values. In my opinion, Discharge prediction in the tidal affected region seems to be useless. I encourage authors to resubmit after carefully adjust their case study, typically in selecting suitable hydrological stations.

Response: Given your comment, we re-work on the experiment using Tan Chau and Chau Doc station pair (~220 km away from the estuary mouth and the entrance of Mekong River Delta), while compared to My Thuan and Can Tho station (as shown in updated Figure 2 with Tan Chau and Chau Doc station time series). The result indicates the impact of backwater effect on the surface runoff due to landward ocean tidal propagation accounts for 9-11% errors (Table 3 in this manuscript) in terms of RMSE normalized by the peak-to-peak runoff values. The runoff reconstruction result is better when Tan Chau station time series was employed.

To justify the usage of Tan Chau and Chau Doc station pair, we add three paragraphs in the datasets (Section 3.1) as below:

"Regarding to discharge variability, the downstream MRB environment is complicated by geographic setting of Mekong river delta (MRD) and ocean tidal propagation landward from the estuary mouth [60]. The MRD is linked to a complicated river system, where the total discharge of the MRB is regulated by Tonle Sap Lake in Cambodia (e.g. [78-80]) before delivering to the delta and the South China Seas (SCS) through several distributaries associated with the two main branches (i.e., Bassac River and Mekong River). As a consequence, the selection of hydrological stations is a critical task.

Given the geographic setting of the MRD, the hydrological stations in the downstream have to be chosen in order to minimize the effect due to discharge regulated by Tonle Sap Lake and ocean tides close to estuary. Though Chaktomuk station is ~300 km away from the estuary mouth, it is at the intersection of several main distributaries. When river flows from several main distributaries meet, the overall temporal pattern of discharge of the main branch of the MRB would be altered. Kratie and Stung Treng stations are too far away from the estuary mouth.

         Given the above criteria, Tan Chau and Chau Doc, located at the MRD entrance with ~220 km away from the estuary mouth, would be the most appropriate hydrological stations that should have a lesser influence by the ocean tidal propagation effect (Figure 1). Note that semi-diurnal (i.e., half-daily period) ocean tides are dominant in SCS, the ocean tidal effect should be mitigated by monthly averaging process. Therefore, both the above two stations were chosen in this study. The Can Tho and My Thuan stations, closest to the estuary mouth, were also chosen for assessing an impact of backwater effect on the surface runoff estimation due to landward ocean tidal propagation."

References:

60. Anthony, E.J.; Brunier, G.; Besset, M.; Goichot, M.; Dussouillez, P.; Nguyen, V.L. Linking rapid erosion of the Mekong river delta to human activities. Sci Rep-UK.2015, 5, 14745.

78. Kummu, M.; Tes, S.; Yin, S.; Adamson, P.; Józsa, J.; Koponen, J.; Richey, J.; Sarkkula, J. Water balance analysis for the Tonle Sap Lake–floodplain system. Hydrol Process.2014, 28(4), 1722-1733.

79. Tangdamrongsub, N.; Ditmar, P.G.; Steele-Dunne, S.C.; Gunter, B.C.;Sutanudjaja, E.H. Assessing total water storage and identifying flood events over Tonlé Sap basin in Cambodia using GRACE and MODIS satellite observations combined with hydrological models. Remote Sens. Environ.2016,181, 162-173.

80. Frappart, F.;Biancamaria, S.;Normandin, C.;Blarel, F.;Bourrel, L.;Aumont, M.; ... Darrozes, J. Influence of recent climatic events on the surface water storage of the Tonle Sap Lake. Sci. Total Environ.2018,636, 1520-1533.

Some other comments

(1) Line 93. The author mentioned WD and Runoff Q. Please use either WD or Q for

consistency in the entire paper.

Response: Many previous studies directly correlated to river water discharge (WD) for WD estimation, except that WD should be in form of surface runoff (R) for consistency with the unit of precipitation, evapotranspiration, and water storage when water balance equation is employed for estimating R. Therefore, from the last few introductory paragraphs to the end of the manuscript, we consistently used R for the entire paper accordingly since our manuscript is partly based on water balance equation.

(2) Line 106-110. Please rephrase for a clarification.

Response: Yes, we rephrase the explanation and clarification sentences from line 91 to 110 as below:

"The Gravity Recovery and Climate Experiment (GRACE) [41] is another active RS technique that observes large-scale time-variable gravity changes for inferring monthly terrestrial water storage (TWS) fluctuations (e.g., [42,43]). GRACE TWS is useful to capture large-scale seasonal surface water changes, which in turn allows monitoring global hydrological cycle and its extremes (e.g., [44,45]). Since WD is a power function of GRACE TWS (S) (e.g., [46,47]), WD can be inferred from S. Another approach to infer WD is based on water balance equation (i.e., ) [48-50], where WD is in the form of surface runoff (R) to be consistent with the unit of precipitation (P), evapotranspiration (ET), and water storage change ().  Besides, S and ΔS are useful to infer precipitation (P) [e.g., 51] and evapotranspiration (ET) (e.g., [52,53]).

The aforementioned hydrological variables (i.e., P, ET, S, and R) are the four water balance components, in which P, ET, and S can be obtained by Tropical Rainfall Measuring Mission (TRMM), MODIS, and GRACE, respectively. These variables are referred to as remotely-sensed hydrological variables (RSHV), while R can only be indirectly inferred. In essence, remotely-sensed P, ET, and S should have a direct causal relationship with R. Their standardized forms, which can be obtained through the subtraction of the data time series from the corresponding mean divided by corresponding standard deviation, should make their averaged time series more representative of regional characteristics [54], in particular across regions with large differences in means and variances [55,56], and hence, reducing the systematic environmental influences [57]. RSHV and their standardized forms are assumedly better at capturing the variations of R and standardized R, respectively, through direct correlative analysis, though standardized hydrological variables has been traditionally served for drought characterization [58,59]. Since R inferred by the water balance equation is achieved through the subtraction among RSHV, systematic environmental influences should be reduced, and thus, expecting to outperform the correlative analysis of each individual RSHV, not to mention the standardization of water balance equation (i.e., ). "

References:

54. Ferreira, V. G., Montecino, H. C., Ndehedehe, C. E., Heck, B., Gong, Z., de Freitas, S. R. C., & Westerhaus, M. (2018). Space-based observations of crustal deflections for drought characterization in Brazil. Science of The Total Environment644, 256-273.

55. Jones, P. D., & Hulme, M. (1996). Calculating regional climatic time series for temperature and precipitation: methods and illustrations. International journal of climatology16(4), 361-377.

56. Li, B., & Rodell, M. (2015). Evaluation of a model-based groundwater drought indicator in the conterminous US. Journal of Hydrology526, 78-88.

(3) Line 115-118. Please add a citation to justify this statement.

Response: We completely revised based on other reviewers' comment as below:

"The downstream MRB is crucial to food (e.g. fish and agriculture products [60]) and water (e.g. [61]) security of Southeast Asia that makes it an attractive geographic study region. While it has been recently undergoing massive hydropower development since 1990s with an aim to regulating R during severe flood and drought period [62,63], the impact of alternation of the upstream R on the downstream MRB is insignificant at a seasonal and annual scale [e.g. 63-66]. Therefore, the hydrology of the MRB should remain dominant by the nature of alternating wet and dry seasons due to monsoons and its geographic setting. This permits us to further investigate the feasibility of using RSHV, their standardized forms as well as water-balance representations (WBR) obtained from the upstream MRB to reconstruct time series of R in the downstream MRB at a monthly temporal scale, for the reason that the hydrology of the upstream MRB shares similarity to that in the downstream MRB except under ENSO influence [67]. The commonly available RSD (i.e., NDVI [68] and LST [69]) are served as baseline results for comparison. The reconstructed relationship is externally validated by estimating the time series of R at other locations in the downstream MRB with in situ station time series used for performance assessment."

(4) Line 169. Figure 2. Please clarify unit of water discharge (mm in monthly, yearly)?. Since Discharge is often in the form of m 3 /s. I think authors should have a paragraph in method part to introduce the way to convert water discharge from m 3 /s to mm.

Response: Yes, we introduce the method of conversion before Figure 2 as below:

" To convert the observed WD (in m3/s) into daily Q rate (in mm/day) per unit area over MRB, the observed WD is divided by total MRB drainage area (i.e. 795,000 km2) to generate daily Q rate. Thus, monthly Q rate (in mm/month) for MRB was computed as the sum of the daily Q rate."

(5) Line 199-200. TRMM data used for SPI calculation seems too short (i.e., nine

years). I suggest author download TRMM dataset for its-era (1998 to present) then

calculate the SPI values for that period but extract only values for the given study

time (2005 – 2013).

Response: Yes, we re-calculated the SPI based on your above suggestion and revised the sentences accordingly. We revised the all related correlation coefficient in respective tables, though the result is not improving much.

(6) Line 242. Please add a citation to justify this statement.

Response: Done.

Reviewer 3 Report

this is a quite clear and useful paper, linking upstream hydrological variables
to downstream river discharge in the Mekong basin

It deserves publication on Remote Rensing and I have no major objection to its rapid publication,
after some issues are fixed:

- the paper should acknowledge previous study adopting a similar procedure (published
in the same journal) i.e. Zampieri et al. (2018). This will add robustness to the
present study because also Zampieri et al. find useful standardizing hydrological variables
for impact studies and the defined the Standardized River Discharge Index. The authors
should also homogeneize their namings as they are actually using the same index as
Zampieri et al. (2018)

- the discussion of the outcomes is questionable on a specific point. The authors claim
that the best skills are obtained with he standardized precipitation index (SPI) from TRMM
with respect to SETI (standardized modis ET) and SI (Grace water storage), but the correlations
with the downstream water discharge are almost the same i.e. 0.911, 0.910 and 0.910, respectively.
I think the right message to convey is that standardization leads to overall better correlations
with respect to the original non-standardized variables, with smaller dependency on the specific
variable (TRMM, MODIS, GRACE).
This is even more evideng in the "external assessment", where the correlations with the downstream
river discharge with the non-standardized variables are 0.76, 075 and 0.72, while the correlations
with the standardized index are equal to 0.87 for each variable.
So I would remove that "the highest NCC is obtained with SPI" in the Concclusions and rephrase
the paragraph so stress just the effect of standardization.

I like very much the fact that including a simple water balance model further increments the
model skills. This is quite a consisten proof of robustness of this study and maybe should
be stressed more.

minor comments:

line 50: "regional hydrological extremes that cause unpredictable losses in agriculture
and economy" -> here Zampieri et al. (2018) should be cited, also because there are not
so many references that are so specific on this field.

somewhere in the Introduction I would add a paragraph stating why Mekong river discharge
is important (agriculture, hydroelectricity, industry, drinking water, ?). This is not
obvious of a general reader.

line 97: "allows us to monitor" -> "allows monitoring"

line 146: "it covers with an area" -> "it covers an area of"

line 149 and later: the authors refer to ENSO as a manifestation of climate change,
but I think they are actually referring to "climate variability" instead. For a
climatologist, climate change is connected to global warming, not to inter-annual
variability

lines 313 - 318: explain better which discrepancies are characterizing the listed
years and why they should be linked to ENSO; add additional analysis of precipitation
downstream, if needed; otherwise, remove this imprecise considerations.

line 325: "abnornal anomalies" what does it mean? all anomalies are abnormal, and
standardization should enphasize anomalies, not reduce them; maybe better to remove
this qualitatiove and unclear consideration (or explain better).

lines 332-333: SPI is not performing better, as a difference with the other indexes
is not significant. Moreover, "climate change" should be "climate variability";
finally, how can the authors state that precipitaiton is more sensitive to climate
change/variability than the other indexes from their statistical analysis?
these considerations are wrong and should be removed. The real result is that
standardization works well for all variables considered.

this paper uses a lot of acronynims compromising readability. I suggest repreating
the acronyms definitions in the Conclusions and in the Figure Captions to ease
superficial readers. I'm not sure the journal likes acronims in the abstract.

Author Response

this is a quite clear and useful paper, linking upstream hydrological variables 
to downstream river discharge in the Mekong basin

Response: Thank you for your comment.

It deserves publication on Remote Rensing and I have no major objection to its rapid publication, 
after some issues are fixed:

- the paper should acknowledge previous study adopting a similar procedure (published in the same journal) i.e. Zampieri et al. (2018). This will add robustness to the present study because also Zampieri et al. find useful standardizing hydrological variables for impact studies and the defined the Standardized River Discharge Index. The authors should also homogeneize their namings as they are actually using the same index as Zampieri et al. (2018)

Response: Yes, we cited this paper in the runoff standardization process. We also change it accordingly throughout the manuscript.

- the discussion of the outcomes is questionable on a specific point. The authors claim 
that the best skills are obtained with he standardized precipitation index (SPI) from TRMM 
with respect to SETI (standardized modis ET) and SI (Grace water storage), but the correlations
with the downstream water discharge are almost the same i.e. 0.911, 0.910 and 0.910, respectively. 
I think the right message to convey is that standardization leads to overall better correlations 
with respect to the original non-standardized variables, with smaller dependency on the specific
variable (TRMM, MODIS, GRACE).
This is even more evideng in the "external assessment", where the correlations with the downstream
river discharge with the non-standardized variables are 0.76, 075 and 0.72, while the correlations
with the standardized index are equal to 0.87 for each variable.
So I would remove that "the highest NCC is obtained with SPI" in the Concclusions and rephrase
the paragraph so stress just the effect of standardization.

Response: Yes, we removed the statement of “the highest PCC”, etc., and emphasize the importance of standardization accordingly throughout the manuscript.

I like very much the fact that including a simple water balance model further increments the 
model skills. This is quite a consisten proof of robustness of this study and maybe should
be stressed more.

Response: Thank you. We revised accordingly.

minor comments:

line 50: "regional hydrological extremes that cause unpredictable losses in agriculture and economy" -> here Zampieri et al. (2018) should be cited, also because there are not so many references that are so specific on this field.

Response: Yes, we cited accordingly.

somewhere in the Introduction I would add a paragraph stating why Mekong river discharge is important (agriculture, hydroelectricity, industry, drinking water, ?). This is not obvious of a general reader.

Response: Yes, we revised accordingly in the 2nd-to-last paragraph, which is shown as below:

" The downstream MRB is crucial to food (e.g. fish and agriculture products [60]) and water (e.g. [61]) security of Southeast Asia that makes it an attractive geographic study region. While it has been recently undergoing massive hydropower development since 1990s with an aim to regulating R during severe flood and drought period [62,63], the impact of alternation of the upstream R on the downstream MRB is insignificant at a seasonal and annual scale [e.g. 63-66]. Therefore, the hydrology of the MRB should remain dominant by the nature of alternating wet and dry seasons due to monsoons and its geographic setting."

References:

60. Anthony, E.J.; Brunier, G.; Besset, M.; Goichot, M.; Dussouillez, P.; Nguyen, V.L. Linking rapid erosion of the Mekong river delta to human activities. Sci Rep.2015,5, 14745.

61. Jacobs, J.W. The Mekong river commission: transboundary water resources planning and regional security. Geogr J.2002,168(4), 354-364.

62. Ziv, G.; Baran, E.; Nam, S.; Rodríguez-Iturbe, I.; Levin, S. A. Trading-off fish biodiversity, food security, and hydropower in the Mekong River Basin. P. Natl Acad. Sci.2012,109(15), 5609-5614.

63. Li, X.; Liu, J.P.; Saito, Y.; Nguyen, V.L. Recent evolution of the Mekong Delta and the impacts of dams. Earth-sci Rev.2017,175, 1-17.

64. Li, S.; He, D. Water level response to hydropower development in the upper Mekong River. AMBIO: A Journal of the Human Environment.2008,37(3), 170-177.

65. Han, Z.; Long, D.; Fang, Y.; Hou, A.; Hong, Y. Impacts of climate change and human activities on the flow regime of the dammed Lancang River in Southwest China. J. Hydrol.2019,570, 96-105.

66. Li, S.J.; He, D.M.; Fu, K.D. The correlations of multi-timescale characteristics of water level processes in Lancang-Mekong river. Chinese Sci Bull.2006,51, 50-58.

line 97: "allows us to monitor" -> "allows monitoring"

Response: Done.

line 146: "it covers with an area" -> "it covers an area of"

Response: Done.

line 149 and later: the authors refer to ENSO as a manifestation of climate change, but I think they are actually referring to "climate variability" instead. For a climatologist, climate change is connected to global warming, not to inter-annual variability

Response: We removed this sentence, as it creates confusion.

lines 313 - 318: explain better which discrepancies are characterizing the listed years and why they should be linked to ENSO; add additional analysis of precipitation downstream, if needed; otherwise, remove this imprecise considerations.

Response: Yes, we removed it.

line 325: "abnornal anomalies" what does it mean? all anomalies are abnormal, and standardization should enphasize anomalies, not reduce them; maybe better to remove this qualitatiove and unclear consideration (or explain better).

Response: Yes, we removed it.

lines 332-333: SPI is not performing better, as a difference with the other indexes is not significant. Moreover, "climate change" should be "climate variability"; finally, how can the authors state that precipitaiton is more sensitive to climate change/variability than the other indexes from their statistical analysis? these considerations are wrong and should be removed. The real result is that standardization works well for all variables considered.

Response: Yes, we removed it.

this paper uses a lot of acronynims compromising readability. I suggest repreating the acronyms definitions in the Conclusions and in the Figure Captions to ease superficial readers. I'm not sure the journal likes acronims in the abstract.

Response: Done.

Reviewer 4 Report

Manuscript entitled “Upstream Remotely-Sensed Hydrological Variables and their Standardization for Downstream Water Discharge Reconstruction and Prediction in the Mekong River Basin.” by Zhou et al.

I appreciate the efforts of the authors in developing the manuscript. This manuscript is well written and easy to follow. The methodology is fine, but it is likely the study area was not chosen carefully.

1.         The Lancang River Basin recently faces the development of hydropower dams. After the construction of Xiaowan, Manwan, Nuozhadu and Jinghong, the hydrological cycle has been modified heavily. There are quite many papers on this topic. The total live storage capacity of dams now accounts for around 35% of the annual flow. The change in storage (delta S), however, was captured by GRACE. In short, the sub-basin is heavily regulated, and this is also reflected in the model. The equation (4) should be only applied for the natural flow (due to the irregular operation of dams)?

Additionally, the authors provided several contradicting information about the region: “Due to MRB’s characteristics, the upper MRB is comparative more stable than that in the downstream MRB” (Lines 115-116), “The temporal pattern of the precipitation of Yunnan province in the upstream basically shares similarity to that in the downstream of MRB” (Lines 153-154) and “Yunnan  Province, a region …. in this study” (Lines 238-240).  

2.         Can Tho and My Thuan stations in the delta are influenced by the tide. I am not so sure about the dataset that the authors obtained from the MRC, but MRC normally provides discharge and water level data measured at 7 AM and 7 PM. The authors may claim that the impact of the tide on monthly water levels could be negligible due to the semi-diurnal regime of the ocean, but the two rivers in the delta link to a complicated river system. Water levels at Can Tho and My Thuan are approximate at the estuaries. Additionally, there is the regulation effect of the Great Lake in Cambodia. There are several papers which confirmed that the influence of dams on the delta is limited (the delta is tidal-dominated). Chiang Saen in Thailand would be a more appropriate station.

3.         The flow at the outlet of the Lancang basin contributes less in the wet and more in the dry season compared to other sub-basin in the Mekong Basin due to the shift of rainfall and storm events in the basin.

Several other minor problems:

I do understand the water balance equation, but I do not so sure about the logic behind its standardized form (line 107). For example, people normally use SPI for drought characterization. It is also noted that SPI is a well-known index so maybe Equations (1) and (2) are not so necessary?

Lines 135 – 138: how about April?

Figure 1: should show the basin boundary. At Phnom Penh in Cambodia, the river flows divide into 3 directions: the Mekong, Bassac and Tonle Rivers.

There are still some grammar mistakes and unclear sentences across the manuscript.

Line 138: “apart from ice and snow melting”? So they also contribute to the flows?

It would be useful if the authors show a map of the Lancang river basin and the GRACE cells belonging to the basin.

I hope the authors find my comments are helpful.

Author Response

I appreciate the efforts of the authors in developing the manuscript. This manuscript is well written and easy to follow. The methodology is fine, but it is likely the study area was not chosen carefully.

1.         The Lancang River Basin recently faces the development of hydropower dams. After the construction of Xiaowan, Manwan, Nuozhadu and Jinghong, the hydrological cycle has been modified heavily. There are quite many papers on this topic. The total live storage capacity of dams now accounts for around 35% of the annual flow. The change in storage (delta S), however, was captured by GRACE. In short, the sub-basin is heavily regulated, and this is also reflected in the model. The equation (4) should be only applied for the natural flow (due to the irregular operation of dams)?

Response:

(i) Yes, we agree that your above statement is true when comparing ground-based hydrological stations for pre-1990s (before dam construction) against post-1990s (after dam construction). Many studies have investigated the dam effect [65]. But most good remotely-sensed data time starting from 2000 to present, in which we used these data time span at "a monthly temporal scale". We realize that dam will alter the flows during the hydrological extremes (i.e. flood and drought) period, in order to ease the flooding and drought, where the magnitude of alteration is apparent for "a few days" during this period [66], and there is no much change at annual and seasonal scale [63,66]. Therefore, we revised one introductory paragraph to accommodate this argument.

" While it has been recently undergoing massive hydropower development since 1990s with an aim to regulating R during severe flood and drought period [62,63], the impact of alternation of the upstream R on the downstream MRB is insignificant at a seasonal and annual scale [e.g. 63-66]. "

62. Ziv, G.; Baran, E.; Nam, S.; Rodríguez-Iturbe, I.; Levin, S. A. Trading-off fish biodiversity, food security, and hydropower in the Mekong River Basin. P. Natl Acad. Sci. 2012, 109(15), 5609-5614.

63. Li, X.; Liu, J.P.; Saito, Y.; Nguyen, V.L. Recent evolution of the Mekong Delta and the impacts of dams. Earth-sci Rev. 2017, 175, 1-17.

64. Li, S.; He, D. Water level response to hydropower development in the upper Mekong River. AMBIO: A Journal of the Human Environment. 2008, 37(3), 170-177.

65. Han, Z.; Long, D.; Fang, Y.; Hou, A.; Hong, Y. Impacts of climate change and human activities on the flow regime of the dammed Lancang River in Southwest China. J. Hydrol. 2019, 570, 96-105.

66. Li, S.J.; He, D.M.; Fu, K.D. The correlations of multi-timescale characteristics of water level processes in Lancang-Mekong river. Chinese Sci Bull. 2006, 51, 50-58.

(ii) Thank you that you already provide a logical answer in your question 2 below that "There are several papers which confirmed that the influence of dams on the delta is limited (the delta is tidal-dominated).". Paragraphs were added and a new experiment was done for the justification of two added stations (i.e. Tan Chau and Chau Doc that is ~220 km away from the estuary mouth and the entrance of Mekong River Delta, when compared to My Thuan and Can tho that is ~100 km away from the estuary mouth) for how much the influence of backwater effect due to landward ocean tidal propagation.

Additionally, the authors provided several contradicting information about the region: “Due to MRB’s characteristics, the upper MRB is comparative more stable than that in the downstream MRB” (Lines 115-116), “The temporal pattern of the precipitation of Yunnan province in the upstream basically shares similarity to that in the downstream of MRB” (Lines 153-154) and “Yunnan  Province, a region …. in this study” (Lines 238-240).  

Response: Thank you for your careful check. We did the correction and deleted contradicting statements in the revised manuscript accordingly. 

2.         Can Tho and My Thuan stations in the delta are influenced by the tide. I am not so sure about the dataset that the authors obtained from the MRC, but MRC normally provides discharge and water level data measured at 7 AM and 7 PM. The authors may claim that the impact of the tide on monthly water levels could be negligible due to the semi-diurnal regime of the ocean, but the two rivers in the delta link to a complicated river system. Water levels at Can Tho and My Thuan are approximate at the estuaries. Additionally, there is the regulation effect of the Great Lake in Cambodia. There are several papers which confirmed that the influence of dams on the delta is limited (the delta is tidal-dominated). Chiang Saen in Thailand would be a more appropriate station.

Response:

(i) The authors purchased the discharge and other data from MRC in October, 2014, in which they provided us with a daily temporal sampling. Scanned agreement is as below:

(ii) Yes, the impact of the tide on monthly water discharge could be mitigated but the influence is still existed. For the new experiment, we conclude the chosen Can Tho and My Thuan stations are subject to ~10% error when compared to that of Tan Chau and Chau Doc which is ~220 km away from the estuary mouth and the entrance of Mekong River Delta. We also write the justification in section 3.1 of the manuscript for the reason of the stations chosen that the reviewer helps the geographic description in the this question 2. These paragraphs are as below:

"Regarding to discharge variability, the downstream MRB environment is complicated by geographic setting of Mekong river delta (MRD) and ocean tidal propagation landward from the estuary mouth [60]. The MRD is linked to a complicated river system, where the total discharge of the MRB is regulated by Tonle Sap Lake in Cambodia (e.g. [78-80]) before delivering to the delta and the South China Seas (SCS) through several distributaries associated with the two main branches (i.e., Bassac River and Mekong River). As a consequence, the selection of hydrological stations is a critical task.

Given the geographic setting of the MRD, the hydrological stations in the downstream have to be chosen in order to minimize the effect due to discharge regulated by Tonle Sap Lake and ocean tides close to estuary. Though Chaktomuk station is ~300 km away from the estuary mouth, it is at the intersection of several main distributaries. When river flows from several main distributaries meet, the overall temporal pattern of discharge of the main branch of the MRB would be altered. Kratie and Stung Treng stations are too far away from the estuary mouth.

Given the above criteria, Tan Chau and Chau Doc, located at the MRD entrance with ~220 km away from the estuary mouth, would be the most appropriate hydrological stations that should have a lesser influence by the ocean tidal propagation effect (Figure 1). Note that semi-diurnal (i.e., half-daily period) ocean tides are dominant in SCS, the ocean tidal effect should be mitigated by monthly averaging process. Therefore, both the above two stations were chosen in this study. The Can Tho and My Thuan stations, closest to the estuary mouth, were also chosen for assessing an impact of backwater effect on the surface runoff estimation due to landward ocean tidal propagation."

References:

60. Anthony, E.J.; Brunier, G.; Besset, M.; Goichot, M.; Dussouillez, P.; Nguyen, V.L. Linking rapid erosion of the Mekong river delta to human activities. Sci Rep-UK.2015, 5, 14745.

78. Kummu, M.; Tes, S.; Yin, S.; Adamson, P.; Józsa, J.; Koponen, J.; Richey, J.; Sarkkula, J. Water balance analysis for the Tonle Sap Lake–floodplain system. Hydrol Process.2014, 28(4), 1722-1733.

79. Tangdamrongsub, N.; Ditmar, P.G.; Steele-Dunne, S.C.; Gunter, B.C.;Sutanudjaja, E.H. Assessing total water storage and identifying flood events over Tonlé Sap basin in Cambodia using GRACE and MODIS satellite observations combined with hydrological models. Remote Sens. Environ.2016,181, 162-173.

80. Frappart, F.;Biancamaria, S.;Normandin, C.;Blarel, F.;Bourrel, L.;Aumont, M.; ... Darrozes, J. Influence of recent climatic events on the surface water storage of the Tonle Sap Lake. Sci. Total Environ.2018,636, 1520-1533.

(iii) Latitude of Chiang Saen in Thailand is 20.27 degree, close to upstream, and it is not at the main Mekong channel; whereas that of Tan Chau is 10.80 degree and ~220km away from the estuary. Therefore, Chiang Saen has not been selected.

3.         The flow at the outlet of the Lancang basin contributes less in the wet and more in the dry season compared to other sub-basin in the Mekong Basin due to the shift of rainfall and storm events in the basin.

Response: Yes.

Several other minor problems:

I do understand the water balance equation, but I do not so sure about the logic behind its standardized form (line 107). For example, people normally use SPI for drought characterization.

Response: Another reviewer also point it out. Yes, SPI and other standardized hydrological variables are normally used for drought characterization, but this study aims to demonstrate the feasibility of the standardization for runoff reconstruction and estimation.

We also revised the texts in the manuscript for clarification on the logic of standardized form and standardized water balance equation as below:

"The aforementioned hydrological variables (i.e., P, ET, S, and R) are the four water balance components, in which P, ET, and S can be obtained by Tropical Rainfall Measuring Mission (TRMM), MODIS, and GRACE, respectively. These variables are referred to as remotely-sensed hydrological variables (RSHV), while R can only be indirectly inferred. In essence, remotely-sensed P, ET, and S should have a direct causal relationship with R. Their standardized forms, which can be obtained through the subtraction of the data time series from the corresponding mean divided by corresponding standard deviation, should make their averaged time series more representative of regional characteristics [54], in particular across regions with large differences in means and variances [55,56], and hence, reducing the systematic environmental influences [57]. RSHV and their standardized forms are assumedly better at capturing the variations of R and standardized R, respectively, through direct correlative analysis, though standardized hydrological variables has been traditionally served for drought characterization [58,59]. Since R inferred by the water balance equation is achieved through the subtraction among RSHV, systematic environmental influences should be reduced, and thus, expecting to outperform the correlative analysis of each individual RSHV, not to mention the standardization of water balance equation (i.e., ). These two forms are referred to as water balance representations."

References:

54. Ferreira, V.G.; Montecino, H.C.; Ndehedehe, C.E.; Heck, B.; Gong, Z.; De, F.S.R.C. et al. Space-based observations of crustal deflections for drought characterization in brazil. Sci. Total Environ.2018,644, 256-273.

55. Jones, P.D.; Hulme, M. Calculating regional climatic time series for temperature and precipitation: methods and illustrations. Int J. Climatol.1996, 16(4), 361-377.

56. Li, B.; Rodell, M. Evaluation of a model-based groundwater drought indicator in the conterminous U.S. J. Hydrol.2015,526, 78-88.

57.He, Q.; Fok, H.; Chen, Q.; Chun, K. Water Level Reconstruction and Prediction Based on Space-Borne Sensors: A Case Study in the Mekong and Yangtze River Basins. Sensors. 2018, 18(9), 3076.

58. Shukla, S.; Wood, A.W. Use of a standardized runoff index for characterizing hydrologic drought. Geo Res Lett.2008,35(2), L02405.

59. Mishra, A.K.; Singh, V.P. A review of drought concepts. J. Hydrol.2010,391(1-2), 202-216.

It is also noted that SPI is a well-known index so maybe Equations (1) and (2) are not so necessary?

Response: Yes, we deleted it accordingly.

Lines 135 – 138: how about April?

Response: We corrected it to "lasts from November to April next year" in line 137.

Figure 1: should show the basin boundary. At Phnom Penh in Cambodia, the river flows divide into 3 directions: the Mekong, Bassac and Tonle Rivers.

Response: Done.

There are still some grammar mistakes and unclear sentences across the manuscript.

Response: Thank you for pointing it out. We checked some grammatical mistakes, unclear sentences, and expression problem, as also noted by other reviewers. We have tried our best to minimize it in this revised manuscript.

Line 138: “apart from ice and snow melting”? So they also contribute to the flows?

Response: Ice and snow melting should contribute very tiny fraction. in order to be conservative, we added this sentence.

It would be useful if the authors show a map of the Lancang river basin and the GRACE cells belonging to the basin.

Response: Done for Lancang river basin; but 1-degree GRACE cells are not visually good. So, we do not add them.

I hope the authors find my comments are helpful.

Response: Very helpful. Thank you for your comment.

Round 2

Reviewer 2 Report

I very appreciate the authors with their workload and novelty in their method. However, I very doubt with the hydrological stations they selected. I suggested two options as below.

[1] If authors still keep predicting runoff. Please clarify discharge measurement as follows:

1.1. Please provide a paragraph describing how discharge is measured at four hydrological stations. If possible, please provide rating curve which calculates discharge from water level.

1.2. Please provide a description statistic of four hydrological stations, including basin area (km2), maximum, minimum, mean, standard deviation of discharge (m3/s); maximum, minimum, mean, standard deviation of water level (m); maximum, minimum, mean, standard deviation of runoff (mm/month).

[2] Author can change to predict the water level. I think water level is the best choice for a simulation at tidal effected region than discharge. In this case, the author should clarify the new contribution, compared to another similar paper on Sensor journal: HE Q., FOK H., CHEN Q., CHUN K. Water Level Reconstruction and Prediction Based on Space-Borne Sensors: A Case Study in the Mekong and Yangtze River Basins. Sensors, 18 (9), 3076, 2018.

Once authors follow one of two suggestions and answer thoughtfully, the manuscript can be accepted.

Other comments.

I believe those four stations author selected are still in the tidal affected region. Kratie (like Datong at Yangtze River) is the last hydrological station in the main Mekong river which is not influenced by tidal. Many hydrological simulations [1,2,3] in Mekong River only worked from Kratie station upward to avoid the tidal dynamics.

[1] ROSSI C., SRINIVASAN R., JIRAYOOT K., LE DUC T., SOUVANNABOUTH P., BINH N., GASSMAN P. Hydrologic evaluation of the lower Mekong river basin with the soil and water assessment tool model. International Agricultural Engineering Journal, 18 (1), 1, 2009.

[2] HOANH CT, JIRAYOOT K, LACOMBE G, V S. Impacts of climate change and development on Mekong flow regimes First assessment-2009; International Water Management Institute: 2010

[3] MOHAMMED I.N., BOLTEN J.D., SRINIVASAN R., LAKSHMI V. Improved Hydrological Decision Support System for the Lower Mekong River Basin Using Satellite-Based Earth Observations. Remote sensing, 10 (6), 2018.

Reviewer 4 Report

Manuscript entitled ” Upstream Remotely-Sensed Hydrological Variables and their Standardization for Downstream Water Discharge Reconstruction and Prediction in the Mekong River Basin” by Zhou et al. submitted to Remote Sensing. The authors handled well my questions; I, however, have several suggestions which might help to improve the manuscript.

1.            Hydropower impacts: among the papers published, Han et al. (2019) did a simulation rather than implemented a historical data analysis. Li et al. (2017) focused on the delta which might be useful for the argument about the delta. Li et al. (2006, 2008) may be obsolete since at that time Manwan and Nuozhadu did not go into operation yet. It may be worth to have a look at the following papers:

Hecht, J. S., Lacombe, G., Arias, M. E., Dang, T. D., & Piman, T. (2018). Hydropower dams of the Mekong River basin: a review of their hydrological impacts. Journal of Hydrology.

Lu, X. X., Li, S., Kummu, M., Padawangi, R., & Wang, J. J. (2014). Observed changes in the water flow at Chiang Saen in the lower Mekong: Impacts of Chinese dams?. Quaternary International, 336, 145-157.

Cochrane, T. A., Arias, M. E., & Piman, T. (2014). Historical impact of water infrastructure on water levels of the Mekong River and the Tonle Sap system. Hydrology and Earth System Sciences, 18(11), 4529-4541.

At Chiang Saen in Thailand, dams already modified the hydrological regime (monthly values) significantly. The authors (References) describe that Chiang Saen is on the main stem of the river (upstream of Chiang Khong and immediately downstream of Jinghong Dam). The authors, however, may claim that dam construction only influence on the last few years' water levels/flows in your data series. Another option is that the authors can remove the year 2013 and 2014 (when Nuozhadu Dam already went online (6 September 2012)) and write a few lines about this.

2.            The influence of tide on water levels in the delta was mentioned in:

Figure 5 in:

Dang, T. D., Cochrane, T. A., & Arias, M. E. (2018). Future hydrological alterations in the Mekong Delta under the impact of water resources development, land subsidence and sea level rise. Journal of Hydrology: Regional Studies, 15, 119-133.

Figure 2 in:

Gugliotta, M., Saito, Y., Nguyen, V. L., Ta, T. K. O., & Tamura, T. (2019). Sediment distribution and depositional processes along the fluvial to marine transition zone of the Mekong River delta, Vietnam. Sedimentology, 66(1), 146-164.

Since Tan Chau and Chau Doc are also influenced by tide (see the two Figures above), the authors can provide a brief paragraph (or a sentence) about uncertainties related to this problem. This is, especially, important because research about the delta is of interests of many other researchers.

3.            The paper below (which the authors also cited in your manuscript) may be helpful when they discuss snow and flows as well.

Räsänen, T. A., & Kummu, M. (2013). Spatiotemporal influences of ENSO on precipitation and flood pulse in the Mekong River Basin. Journal of Hydrology, 476, 154-168.